# THE **PIMMUR** PRINCIPLES: ENSURING VALIDITY IN COLLECTIVE BEHAVIOR OF LLM SOCIETIES

## ABSTRACT

Large Language Models (LLMs) are increasingly used for social simulation, where populations of agents are expected to reproduce human-like collective behavior. However, we find that many recent studies adopt experimental designs that systematically undermine the validity of their claims. From a survey of over 40 papers, we identify six recurring methodological flaws: agents are often homogeneous (**P**rofile), interactions are absent or artificially imposed (**I**nteraction), memory is discarded (**M**emory), prompts tightly control outcomes (**M**inimal-Control), agents can infer the experimental hypothesis (**U**nawareness), and validation relies on simplified theoretical models rather than real-world data (**R**ealism). For instance, GPT-4o and Qwen-3 infer the underlying social-experiments in 52.9% of cases—for example, identifying that their herd behaviors are being tested—despite not being provided with such information, thereby violating the Unawareness principle. We formalize these six requirements as the **PIMMUR** principles and argue they are necessary conditions for credible LLM-based social simulation. To demonstrate their impact, we re-run five representative studies using a framework that enforces **PIMMUR** and find that the reported social phenomena frequently fail to emerge under more rigorous conditions. Our work establishes methodological standards for LLM-based multi-agent research and provides a foundation for more reliable and reproducible claims about "AI societies."

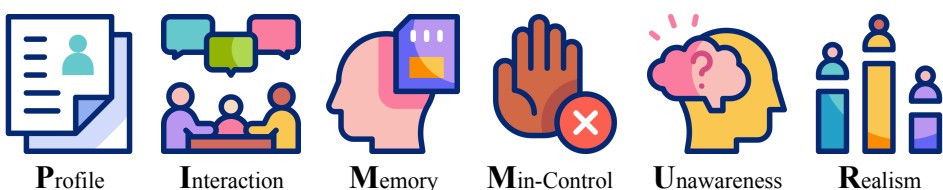

| **P**rofile | **I**nteraction | **M**emory | **M**in-Control | **U**nawareness | **R**ealism |

Figure 1: The **PIMMUR** principles. The first three (**PIM**) focus on micro-level agent designs, while the latter three (**MUR**) focus on macro-level experiment designs.

## 1 INTRODUCTION

Large Language Models (LLMs) have rapidly advanced in their reasoning (Huang et al., 2025a), communication (Tran et al., 2025), and coordination capabilities (Agashe et al., 2025), sparking growing interest in their potential applications within human society, including human digital twins and virtual worlds. One particularly active line of research is LLM-based multi-agent social simulation, where multiple LLM agents are deployed to emulate collective behaviors, institutions, and social phenomena (Mou et al., 2024; Zhang et al., 2024a; Liu et al., 2025a, *inter alia*). The central research question is to examine whether emergent patterns observed in human societies, such as cooperation, polarization, rumor spreading, or network formation, can also arise in societies composed purely of LLM agents. Such simulations promise new methodologies for computational social science and raise the prospect of "AI societies" serving as testbeds for sociological theories.

Despite the enthusiasm, the validity of current LLM-based social simulation studies remains deeply uncertain (Anthis et al., 2025). Existing works often rely on customized frameworks tailored to specific social experiments (Yang et al., 2024; Borah et al., 2025), but methodological choices vary

widely and lack shared standards (Zhou et al., 2024). As a result, it is unclear whether their reported findings reflect genuine emergent social phenomena, or artifacts of flawed experimental design.

In this paper, we critically review the current practices in social simulations, conducting an in-depth analysis of more than 40 papers. Consequently, we identify six recurring methodological pitfalls that undermine the credibility of conclusions: (1) *Homogeneity* – agents lack individuality, often being instantiated from identical LLMs without distinct profiles, backgrounds, or personalities. (2) *Pseudo-Multi-Agent Design* – studies present themselves as multi-agent, yet reduce interaction to single-agent reasoning with injected external signals. (3) *Statelessness* – agents are deprived of memory, preventing them from developing persistent beliefs or evolving social identities. (4) *Goal-Injected Bias* – prompts often encode experimental hypotheses directly, leading to results that are scripted rather than emergent. (5) *Experimenter Visibility Effect* – powerful LLMs can infer the experimental setup itself, introducing systematic biases akin to demand characteristics (Orne, 2017), the Hawthorne effect (Adair, 1984), and social desirability bias (Grimm, 2010). (6) *Model-Model Circularity* – simulations aim to reproduce stylized mathematical models of society rather than grounding themselves in empirical human data, producing circular validations instead of genuine discoveries. Together, these flaws invalidate the target social experiments by ignoring key human-like features, thus producing conclusions that are often unreliable.

To address these issues, we distill the above issues into six methodological principles, *i.e.*, Profile, Interaction, Memory, Minimal-Control, Unawareness, and Realism (**PIMMUR**), that any credible LLM-based social simulation should satisfy. The first three principles focus on micro-level agent design—ensuring that agents function as sufficiently rich analogues of human individuals. The latter three safeguard macro-level experimental design against biases that invalidate emergent outcomes. Adherence to these principles is crucial for producing simulations that are both scientifically interpretable and sociologically meaningful. We find that only four papers satisfy all six principles.

To demonstrate the consequences of violating the principles, we build a new framework that enforces these principles based on SOTOPIA-S4 (Zhou et al., 2025), and re-run five canonical social experiments: (1) fake news propagation (Liu et al., 2024b), (2) social balance (Cisneros-Velarde, 2024), (3) telephone game effect (Liu et al., 2025a), (4) herd effect (Cho et al., 2025), and (5) social network growth (De Marzo et al., 2023). We show that when **PIMMUR** is enforced, conclusions reported in prior work fail to replicate, highlighting the fragility of current methodological standards and conclusions. For example, prior work reported that 56.1% of LLM agents exhibited confirmation bias (Liu et al., 2024b) and that social relationships were balanced in 60.7% of cases (Cisneros-Velarde, 2024). In contrast, our replication yields markedly lower rates of 32.8% and 10.9%, respectively, indicating that LLM prosocial behaviors may have been over-estimated due to violation of *Min-Control* and *Unawareness*. Our contributions include:

1. We propose **PIMMUR** as the first systematic standard for LLM-based social simulation, pointing out current design flaws and offering concrete criteria for credible experimental design.
2. We review more than **40** studies and reveal widespread violations of these basic principles, raising concerns about the reliability and validity of reported findings.
3. We implement a **PIMMUR**-compliant framework and re-run five influential experiments, demonstrating how violations lead to biased, unreliable, or invalid results.

## 2 BACKGROUND: THE EXPERIMENTER VISIBILITY EFFECTS

Real-world social experiments can be biased by several well-documented factors. This section introduces these effects and draws parallels to similar phenomena in AI.

**Demand Characteristics.** This concept refers to cues in a study that lead participants to infer the purpose of the experiment and adjust their behavior accordingly (Orne, 2017). Instead of responding naturally, they may act in ways that align with what they think the researcher expects. Similar effects have been explored in AI community, where models identify superficial cues in questions to select the correct answers, which is often referred to shortcut learning (Li et al., 2025c; Balepur et al., 2024) or Clever Hans effect[1] (McCoy et al., 2019; Shapira et al., 2024; Ullman, 2023). In LLM

---

[1]It originates from a German horse in the early 1900s that seemed to solve math problems but was actually responding to subtle, unconscious cues from its trainer.

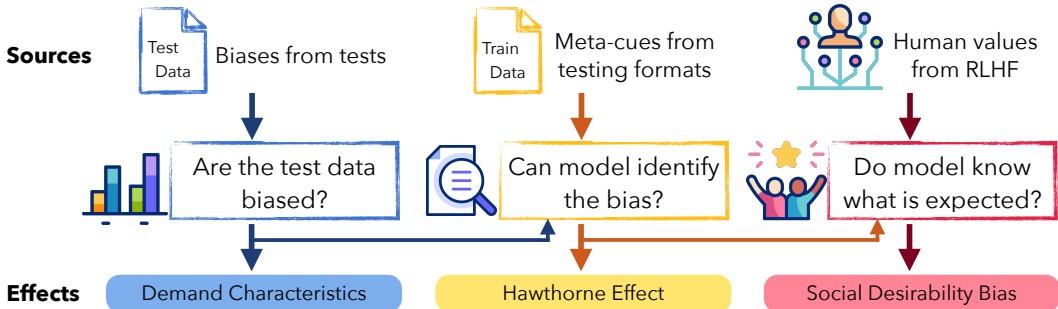

Figure 2: An illustration of how the three experimenter visibility effects interact.

social simulations, overly directive prompts can serve as such cues, causing the model's responses to reflect instructions rather than genuine emergent behavior.

**Hawthorne Effect.** This effect describes the tendency of individuals to change their behavior simply because they are aware of being observed (Adair, 1984), originated from classic workplace studies (Mayo, 2004; Roethlisberger & Dickson, 2003). Recent work in AI community has found that LLMs can know they are being evaluated (Needham et al., 2025; Shao et al., 2025; Nguyen et al., 2025), and they can change their behaviors accordingly. For LLM societies, if the model can infer that it is being tested on a specific social phenomenon (*e.g.*, conformity, cooperation), its behavior may artificially converge toward the "expected" outcome.

**Social Desirability Bias.** This bias occurs when participants present themselves in ways that they believe are socially acceptable rather than truthful, such as overstating prosocial behavior (Grimm, 2010) or downplaying controversial opinions (Nederhof, 1985). Analogously, LLMs often generate outputs that conform to social norms or ethical expectations (Salecha et al., 2024; Lee et al., 2024; Watson et al., 2025) and "resist" to behave socially undesirably (Li et al., 2025b; Huang et al., 2024b). This phenomenon is also linked to sycophancy in LLMs (Fanous et al., 2025; Cheng et al., 2025). This bias can distort simulation results, making AI-based societies appear more cooperative or normative than they actually are.

## 3 THE **PIMMUR** PRINCIPLES

We identify six recurring issues in the existing literature, based on 42 studies (§4), by summarizing their reported advantages and disadvantages.

> **Definition: Profile**
>
> Agents should have distinct backgrounds, preferences, or cognitive styles, such that the system exhibits heterogeneity beyond a homogeneous replication of a single model.

Homogeneity in multi-agent systems can lead to undesirable outcomes such as the echo chamber effect (Reid et al., 2025) and conversation collapse (Pitre et al., 2025; Hegazy, 2024). Recent studies demonstrate that modulating LLMs' personas (Chen et al., 2024; Wang et al., 2024; Huang et al., 2024b; Wang et al., 2025c) or emotional states (Huang et al., 2024a) can influence their behaviors. In contrast, personality is an inherent human attribute rather than an optional modeling choice. To better approximate real-world settings, where individuals' beliefs and preferences diverge, systems should incorporate LLMs with different backbones or distinct backgrounds and preferences. This is essential for studying how specific traits shape collective outcomes, for example, Cho et al. (2025) examine how peers' different education levels influence herd effect.

> **Definition: Interaction**
>
> Agents should influence and respond to one another either directly through messages or indirectly through changing the environment that others can perceive, rather than merely reacting to user-specified statistical summaries.

Many studies claim to focus on multi-agent settings but, in practice, rely on single-agent formulations. For instance, research on herd effects (Cho et al., 2025; Liu et al., 2025a) often embeds aggregated information directly in prompts, such as "Given that 10 others have chosen A, what will you select?" Similarly, in work on social network growth (De Marzo et al., 2023), a single agent is provided with others' friend counts to decide whether to form new connections. While such simplifications make simulation more tractable, their conclusions are difficult to generalize to real-world scenarios, where dynamic interactions among agents or changes to the environments crucially shape individual behaviors (Ran et al., 2025; Huang et al., 2025b;a).

> **Definition: Memory**
>
> Agents should maintain and update persistent internal states across time, allowing information to be internalized, retained, and re-expressed rather than paraphrased statelessly.

Many existing studies evaluate LLMs in single-round settings, overlooking the role of information internalization. For instance, research on the telephone game effect (also known as the rumor chain effect) (Liu et al., 2025a; Perez et al., 2025) typically instructs an LLM to iteratively paraphrase a message. However, this setup diverges from real-world dynamics, where individuals first internalize a message before retransmitting it through their own narratives (Hirst & Echterhoff, 2012). Recent work further emphasizes that such simplifications both misrepresent human cognition and expose limitations in current memory architectures (Wang et al., 2025b).

> **Definition: Minimal-Control**
>
> A simulation should minimize "hints" in both its prompts and systems, instead providing only the essential instructions for perception, action, and communication. This property ensures that observed behaviors are emergent rather than artifacts of researcher-imposed rules.

Prompt engineering has been shown to be effective for a wide range of LLM-based downstream applications (Sahoo et al., 2024; Schulhoff et al., 2024). The same approach is also employed in LLM-based social simulation. For example, to examine belief convergence in fake news propagation, Liu et al. (2024b) explicitly instructed the model to "demonstrate confirmation bias." We argue that while detailed instructions often yield seemingly significant results, such outcomes may primarily reflect prompt design rather than the model's intrinsic behavior. Note that if profiles are intentionally biased to induce certain outcomes (*e.g.*, making most agents highly aggressive in a study of general populations), this would indeed violate Min-Control.

> **Definition: Unawareness**
>
> Agents in the simulation should remain unaware of the experimental hypothesis, design, and evaluation criteria. By preventing meta-awareness of the research setup, this property reduces systematic biases, yielding more reliable behavioral outcomes.

In contrast to *Minimal-Control*, which primarily concerns the researcher's role, *Unawareness* pertains to the models themselves. Even a carefully designed minimal-control instruction may still reflect a classical experimental setup so famous that models can easily infer, as illustrated by Masumori & Ikegami (2025)'s Sugarscape experiment and Cau et al. (2025)'s studies on opinion dynamics. This suggests that models may be "too intelligent" for certain social experiments, as their responses can be influenced by the Hawthorne effect or social desirability bias. We therefore recommend that, prior to conducting LLM-based social experiments, researchers use prompts to test whether models can infer the experimental objective.

> **Definition: Realism**
>
> A simulation should use empirical data from real human societies as references rather than only reproducing simplified theoretical models. This property ensures that emergent behaviors can be meaningfully validated against observed human dynamics.

To analyze complex human societies, social scientists have long relied on simplified mathematical models, such as Heider's balance theory (Heider, 1946) and the Sugarscape experiment (Epstein & Axtell, 1996). In LLM-based social simulations, many studies treat these models as ground truth for evaluating outcomes (De Marzo et al., 2023; Cisneros-Velarde, 2024; Borah et al., 2025). However,

Table 1: Related work on LLM-based multi-agent simulation of different human social phenomena. GPT, Gemini, Gemma, Claude, LLaMA, Qwen, Mistral (including Mixtral), GLM, and Vicuna are denoted by (GPT), (Gemini), (Gemma), (Claude), (LLaMA), (Qwen), (Mistral), (GLM), and (Vicuna) respectively. Simulation goals are explained in Table 3 in §D of the appendix.

| Papers | Simulation Goal | Language Models | P. | I. | Mem. | Min-Ctrl. | U. | R. |
|---|---|---|---|---|---|---|---|---|
| Weng et al. (2025) | Conformity | GPT, Gemma, LLaMA, Qwen | ✗ | ✗ | ✗ | ✓ | ✗ | ✗ |
| Großmann et al. (2025) | Narrative Priming | LLaMA | ✗ | ✗ | ✗ | ✗ | ✓ | ✗ |
| Perez et al. (2025) | Telephone Game | GPT, LLaMA, Mistral | ✗ | ✗ | ✗ | ✗ | ✗ | ✗ |
| De Marzo et al. (2023) | Social Network Growth | GPT | ✗ | ✗ | ✗ | ✓ | ✗ | ✗ |
| Cisneros-Velarde (2024) | Social Balance | LLaMA, Mistral | ✗ | ✗ | ✗ | ✓ | ✗ | ✗ |
| Song et al. (2025) | Trust Formation | LLaMA, Qwen | ✗ | ✗ | ✓ | ✓ | ✓ | ✗ |
| Mittelstädt et al. (2024) | Social Situational Judgments | GPT, Gemini, Claude | ✗ | ✗ | ✗ | ✓ | ✓ | ✓ |
| Li et al. (2025d) | Collective Reasoning | GPT, Gemini, LLaMA, Qwen | ✗ | ✓ | ✗ | ✓ | ✗ | ✗ |
| Cau et al. (2025) | Opinion Dynamic | LLaMA, Mistral | ✗ | ✓ | ✗ | ✓ | ✓ | ✗ |
| Cho et al. (2025) | Herd Effect | GPT | ✓ | ✗ | ✗ | ✓ | ✗ | ✗ |
| Li et al. (2024) | Fake News Propagation | GPT | ✓ | ✗ | ✗ | ✗ | ✓ | ✗ |
| Zhang et al. (2024a) | Collaboration | GPT, LLaMA, Qwen, Mistral | ✓ | ✗ | ✗ | ✗ | ✓ | ✓ |
| Huang et al. (2024c) | Profile Consistency | GPT, Gemini, Claude, LLaMA, Qwen, Mistral, GLM | ✓ | ✗ | ✗ | ✓ | ✗ | ✗ |
| Borah et al. (2025) | Belief Congruence | GPT, LLaMA, Qwen | ✗ | ✓ | ✓ | ✗ | ✗ | ✗ |
| Piatti et al. (2024) | Sustainable Cooperation | GPT, LLaMA, Qwen, Mistral | ✗ | ✓ | ✓ | ✓ | ✗ | ✓ |
| Masumori & Ikegami (2025) | Sugarscape | GPT, Claude | ✗ | ✓ | ✓ | ✓ | ✗ | ✗ |
| Yuzhe et al. (2026) | Financial Market | GPT | ✓ | ✓ | ✗ | ✗ | ✓ | ✓ |
| Liu et al. (2025a) | Prosocial Irrationality | GPT, Gemini, Claude, Mistral | ✓ | ✗ | ✗ | ✗ | ✗ | ✗ |
| Chuang et al. (2024) | Wisdom of Partisan Crowds | GPT, Vicuna | ✓ | ✗ | ✓ | ✓ | ✗ | ✗ |
| Zhang et al. (2024b) | Election | GPT, Gemini, Claude | ✓ | ✗ | ✓ | ✓ | ✗ | ✓ |
| Liu et al. (2024c) | Fake News Evolution | GPT | ✓ | ✓ | ✓ | ✗ | ✓ | ✓ |
| Ren et al. (2024) | Social Norm | GPT | ✓ | ✓ | ✓ | ✗ | ✗ | ✓ |
| Hou et al. (2025) | Vaccine Hesitancy | LLaMA | ✓ | ✓ | ✓ | ✗ | ✗ | ✗ |
| Hua et al. (2023) | World War | GPT, Claude | ✓ | ✓ | ✓ | ✓ | ✗ | ✗ |
| Mou et al. (2025) | Social Intelligence | GPT, LLaMA, Qwen, Mistral | ✓ | ✓ | ✓ | ✗ | ✓ | ✗ |
| Liu et al. (2024b) | Fake News Propagation | GPT | ✓ | ✓ | ✓ | ✗ | ✓ | ✗ |
| Tomašević et al. (2025) | Operational Validity | LLaMA | ✓ | ✓ | ✗ | ✓ | ✓ | ✓ |
| Zhang et al. (2025) | Trending Topic | GLM | ✓ | ✓ | ✓ | ✓ | ✓ | ✓ |
| Mou et al. (2024) | Echo Chambers | GPT | ✓ | ✓ | ✓ | ✗ | ✓ | ✓ |
| Yang et al. (2024) | Polarization | LLaMA | ✓ | ✓ | ✓ | ✓ | ✓ | ✓ |
| Touzel et al. (2024) | Social Manipulation | GPT | ✓ | ✓ | ✓ | ✗ | ✓ | ✓ |
| Park et al. (2023) | Agent Coordination | GPT | ✓ | ✓ | ✓ | ✓ | ✓ | ✓ |
| Wang et al. (2025a) | Cultural Dissemination | GPT | ✓ | ✓ | ✓ | ✗ | ✗ | ✗ |
| Piao et al. (2025) | Polarization | GPT | ✓ | ✓ | ✗ | ✗ | ✓ | ✗ |

because the ultimate objective is to approximate real-world human behavior, alignment with overly simplified models provides limited insight (Larooij & Törnberg, 2025). Conversely, whether AI societies reproduce the phenomena of mathematical theories should not be regarded as a measure of AI social intelligence. Models not grounded to human data should not serve as primary references.

# 4 LITERATURE REVIEW IN THE LENS OF **PIMMUR**

**Overview.** To illustrate the severity of these issues in existing LLM-based social simulation research, we review 42 papers in this section. We focus on studies that employ LLMs to simulate human social phenomena using multi-agent systems, excluding those using LLM agents solely for downstream tasks such as coding or mathematical problem solving. Excluding eight studies (Zheng & Tang, 2024; Orlando et al., 2025; Tang et al., 2025; Liu et al., 2025b; Gu et al., 2025; Liu et al., 2024a; Gao et al., 2023; He et al., 2023) that did not release their frameworks at the time of our study (and thus could not be evaluated for their design), Table 1 summarizes the 34 reviewed papers,

Table 2: Results of using LLM to evaluate instructions of papers from Table 1. The numbers show the percentages of ✗. Models are GPT-4o, Gemini-2.5, Claude-4, LLaMA-4, and Qwen-3.

(a) *Unawareness* evaluation. ✗ denotes models can infer correctly, indicating a violence of this principle.

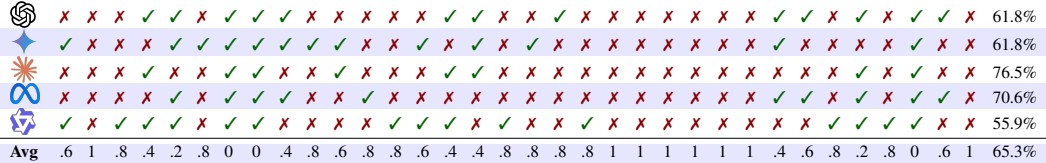

| Language Models | Weng et al. (2025) | Großmann et al. (2025) | Perez et al. (2025) | De Marzo et al. (2023) | Cisneros-Velarde (2024) | Song et al. (2025) | Mittelstädt et al. (2024) | Li et al. (2025d) | Cau et al. (2025) | Cho et al. (2025) | Li et al. (2024) | Zhang et al. (2024a) | Huang et al. (2024c) | Borah et al. (2025) | Piatti et al. (2024) | Masumori & Ikegami (2025) | Yuzhe et al. (2026) | Liu et al. (2025a) | Chuang et al. (2024) | Zhang et al. (2024b) | Liu et al. (2024c) | Ren et al. (2024) | Hou et al. (2025) | Hua et al. (2023) | Mou et al. (2025) | Liu et al. (2024b) | Tomašević et al. (2025) | Zhang et al. (2025) | Mou et al. (2024) | Yang et al. (2024) | Touzel et al. (2024) | Park et al. (2023) | Wang et al. (2025a) | Piao et al. (2025) | Average |
|---|---|---|---|---|---|---|---|---|---|---|---|---|---|---|---|---|---|---|---|---|---|---|---|---|---|---|---|---|---|---|---|---|---|---|---|
| GPT-4o | ✗ | ✓ | ✓ | ✗ | ✗ | ✗ | ✗ | ✓ | ✗ | ✗ | ✓ | ✓ | ✓ | ✗ | ✗ | ✗ | ✓ | ✗ | ✗ | ✗ | ✗ | ✗ | ✗ | ✓ | ✓ | ✗ | ✓ | ✓ | ✓ | ✓ | ✓ | ✓ | ✓ | ✗ | 52.9% |
| Gemini-2.5 | ✗ | ✗ | ✓ | ✓ | ✗ | ✗ | ✓ | ✓ | ✗ | ✗ | ✓ | ✓ | ✓ | ✗ | ✗ | ✗ | ✓ | ✗ | ✗ | ✗ | ✓ | ✗ | ✗ | ✓ | ✓ | ✗ | ✓ | ✓ | ✓ | ✓ | ✓ | ✓ | ✓ | ✗ | 47.1% |
| Claude-4 | ✗ | ✗ | ✓ | ✓ | ✗ | ✗ | ✓ | ✗ | ✓ | ✗ | ✓ | ✗ | ✓ | ✗ | ✗ | ✗ | ✓ | ✗ | ✗ | ✗ | ✗ | ✗ | ✗ | ✓ | ✓ | ✓ | ✓ | ✓ | ✓ | ✓ | ✓ | ✓ | ✓ | ✗ | 47.1% |
| LLaMA-4 | ✗ | ✓ | ✓ | ✗ | ✗ | ✗ | ✓ | ✓ | ✓ | ✗ | ✓ | ✗ | ✓ | ✗ | ✗ | ✗ | ✓ | ✗ | ✓ | ✗ | ✓ | ✓ | ✓ | ✓ | ✓ | ✓ | ✓ | ✓ | ✗ | ✓ | ✓ | ✓ | ✓ | ✗ | 38.2% |
| Qwen-3 | ✗ | ✗ | ✓ | ✗ | ✗ | ✗ | ✗ | ✓ | ✗ | ✗ | ✓ | ✓ | ✓ | ✗ | ✗ | ✗ | ✓ | ✗ | ✗ | ✗ | ✗ | ✗ | ✗ | ✓ | ✓ | ✓ | ✓ | ✓ | ✓ | ✓ | ✓ | ✓ | ✓ | ✗ | 52.9% |
| **Avg** | 1 | .6 | 0 | .6 | 1 | 1 | .4 | .2 | .6 | 1 | 0 | .4 | 0 | 1 | 1 | 1 | 0 | 1 | .6 | 1 | .6 | .8 | .8 | 0 | 0 | .4 | 0 | 0 | .2 | 0 | 0 | 0 | 0 | 1 | 47.6% |

(b) *Minimal-Control* evaluation. ✗ denotes extra instructions detected, indicating a violence of this principle.

| Language Models | Weng | Großmann | Perez | De Marzo | Cisneros | Song | Mittelstädt | Li 2025d | Cau | Cho | Li 2024 | Zhang 2024a | Huang | Borah | Piatti | Masumori | Yuzhe | Liu 2025a | Chuang | Zhang 2024b | Liu 2024c | Ren | Hou | Hua | Mou 2025 | Liu 2024b | Tomašević | Zhang 2025 | Mou 2024 | Yang | Touzel | Park | Wang | Piao | Average |
|---|---|---|---|---|---|---|---|---|---|---|---|---|---|---|---|---|---|---|---|---|---|---|---|---|---|---|---|---|---|---|---|---|---|---|---|
| GPT-4o | ✓ | ✗ | ✗ | ✓ | ✓ | ✗ | ✓ | ✓ | ✗ | ✗ | ✓ | ✗ | ✗ | ✓ | ✓ | ✓ | ✗ | ✗ | ✗ | ✗ | ✗ | ✗ | ✗ | ✗ | ✗ | ✗ | ✗ | ✓ | ✗ | ✓ | ✗ | ✓ | ✓ | ✗ | 61.8% |
| Gemini-2.5 | ✓ | ✗ | ✗ | ✓ | ✓ | ✗ | ✓ | ✓ | ✓ | ✗ | ✓ | ✗ | ✗ | ✓ | ✓ | ✗ | ✗ | ✗ | ✗ | ✗ | ✗ | ✗ | ✗ | ✗ | ✗ | ✗ | ✓ | ✓ | ✗ | ✗ | ✗ | ✓ | ✓ | ✗ | 61.8% |
| Claude-4 | ✗ | ✗ | ✗ | ✗ | ✓ | ✗ | ✓ | ✓ | ✓ | ✗ | ✗ | ✗ | ✗ | ✗ | ✗ | ✓ | ✗ | ✗ | ✗ | ✗ | ✗ | ✗ | ✗ | ✗ | ✗ | ✗ | ✓ | ✗ | ✗ | ✓ | ✗ | ✓ | ✗ | ✗ | 76.5% |
| LLaMA-4 | ✗ | ✗ | ✗ | ✓ | ✓ | ✗ | ✓ | ✓ | ✓ | ✗ | ✗ | ✗ | ✗ | ✗ | ✓ | ✓ | ✗ | ✗ | ✗ | ✗ | ✗ | ✗ | ✗ | ✗ | ✗ | ✗ | ✓ | ✗ | ✗ | ✓ | ✗ | ✓ | ✗ | ✗ | 70.6% |
| Qwen-3 | ✗ | ✗ | ✓ | ✗ | ✗ | ✓ | ✓ | ✓ | ✗ | ✓ | ✗ | ✓ | ✓ | ✗ | ✗ | ✗ | ✓ | ✓ | ✓ | ✓ | ✗ | ✗ | ✗ | ✗ | ✗ | ✗ | ✗ | ✗ | ✓ | ✓ | ✓ | ✓ | ✗ | ✗ | 55.9% |
| **Avg** | .6 | 1 | .8 | .4 | .2 | .8 | 0 | 0 | .4 | .8 | .6 | .8 | .8 | .6 | .4 | .4 | .8 | .8 | .8 | .8 | 1 | 1 | 1 | 1 | 1 | 1 | .4 | .6 | .8 | .2 | .8 | 0 | .6 | 1 | 65.3% |

including their simulation goals, selected models, and compliance with the **PIMMUR** principles. The assessment of compliance is determined by the authors through discussion, based on the definitions provided in §3. Only three papers (Zhang et al., 2025; Yang et al., 2024; Park et al., 2023) fully satisfy all principles. These works focus on large-scale, real-world simulations with diverse agent action spaces, preventing individuals from inferring the experimental objective. The review highlights that both the principles and their implications, such as the risk of invalidated conclusions, have not received sufficient attention in the community.

**Evaluating Unawareness** Except for *Min-Control* and *Unawareness*, all principles can be evaluated straightforwardly. By definition, *Unawareness* assesses whether LLMs possess the knowledge of the overall experimental design; the most direct evaluation is therefore to query the models explicitly. We devise a prompt (Fig. 10) that requires models to infer the simulation goal solely from the given instructions. Using instructions from all 34 papers, we query five frontier models: GPT-4o-2024-11-20 (Hurst et al., 2024), Gemini-2.5-Flash-05-20 (Kavukcuoglu, 2025), Claude-4.0-Sonnet (Anthropic, 2025), LLaMA-4-Scout (Meta, 2025), and Qwen-3-235B-A22B (Yang et al., 2025). As summarized in Table 2a, stronger models such as GPT-4o and Qwen-3 correctly inferred the simulation goal in 52.9%% of cases. Compared to humans, LLMs more accurately recognize experimental settings and expected outcomes in cases where human annotators possess limited domain knowledge. This suggests that frontier models may be too smart to serve as subjects in social experiments, illustrating a "curse of knowledge" (Li et al., 2025a).

**Evaluating Minimal-Control** We design a second prompt (Fig. 11) that asks models to act as social psychology experts and determine whether the instructions contain unnecessary steering components. As shown in Table 2b, the five LLMs mark 64.4% instructions as over-control, suggesting that many conclusions in prior studies may be artifacts of researcher-imposed rules. As discussed in §2, such experimenter visibility effects systematically bias model behavior and consequently bias the conclusions. Compared to human judgments in Table 1, LLMs tend to be overly strict, frequently labeling neutral instructions as instances of over-control. This discrepancy likely arises because LLMs pay too much attention on instructions and simulation goals, whereas human annotators also consider the full design. For example, LLMs wrongly flag assigned traits (*e.g.*, Openness, Extroversion) as potentially steering, which serve to represent a diverse population sampled according to real-world distributions, thereby enhancing ecological validity rather than undermining neutrality.

## 5 REPRODUCING SOCIAL SIMULATIONS WITH **PIMMUR**

To demonstrate the differences in conclusions under compliance and violation of **PIMMUR**, we re-run five social experiments that violate our principles: (1) fake news propagation (Liu et al., 2024b), (2) social balance (Cisneros-Velarde, 2024), (3) telephone game effect (Liu et al., 2025a), (4) herd effect (Cho et al., 2025), and (5) social network growth (De Marzo et al., 2023).

We implement our **PIMMUR**-compliant framework as follows (more details are included in §B): *(1) Heterogeneous profiles:* Each agent is initialized with a Big Five personality profile (John et al., 1991) and a short life story generated according to that personality. *(2) Persistent memory:* Agents are equipped with a memory module for information storage and retrieval. We implement two variants: a lightweight version that directly store recent interactions, and an advanced design using reflection to iteratively manage (Park et al., 2023). *(3) Communication protocol.* Our framework supports different network structure, including chain-based networks (*e.g.*, telephone game), dynamic graphs (*e.g.*, network growth), and fully connected graphs (*e.g.*, fake news, social balance, or herd effect). Visibility can be toggled between private messages and broadcasting. In our simulations, all agents take turns communicating in a round-robin fashion rather than free form discussion. *(4) Experimental settings.* Agents are given only the minimal information necessary to complete their tasks, without disclosing goals, motivations, or expected outcomes. Empirically, we ensure all the experimental instructions pass GPT-4o's *Unawareness* and *Minimal-Control* check.

### 5.1 BELIEF CONVERGENCE (FAKE NEWS PROPAGATION)

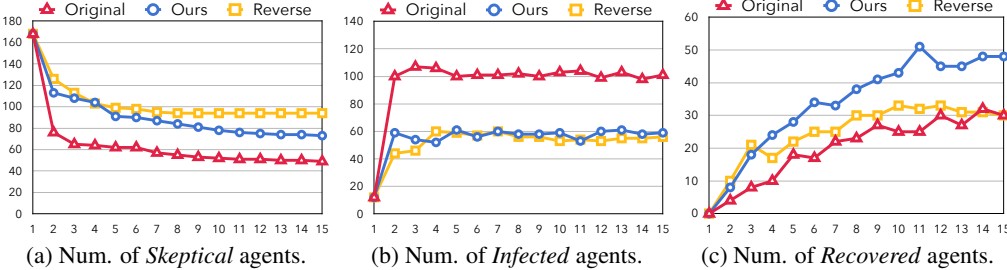

(a) Num. of *Skeptical* agents.  (b) Num. of *Infected* agents.  (c) Num. of *Recovered* agents.

Figure 3: Dynamics of agents in different states (skeptical, infected, recovered) across rounds.

Liu et al. (2024b) investigate whether agents accept fake news and exhibit confirmation bias. At the beginning of their simulation, an agent is seeded with fake news. In each round, agents interact with randomly selected peers, reflect on the news they have encountered, and decide whether to believe it. At the end of each round, we record three populations: (1) skeptical agents who have never believed the fake news, (2) infected agents who currently believe it, and (3) recovered agents who once believed it but no longer do. A central limitation of their setup is over-control in the instructions: *"... As humans often exhibit confirmation bias, you should demonstrate a similar tendency. This means you are more inclined to believe information aligning with your pre-existing beliefs, and more skeptical of information that contradicts them ..."* This directive violates the principle of **Min-Control** and biases agents toward susceptibility to misinformation. In contrast, we first remove this line and then additionally introduce a reversed instruction (*"you are more inclined to believe information that contradicts your pre-existing beliefs, and more skeptical of information aligning with them"*). As shown in Fig. 3, the number of infected agents drops from 56.11% to 32.78%. Many report in their reasoning that they "need more information and validation" before believing the fake news, indicating that LLM agents show less confirmation bias shown previously. Furthermore, when the fake news concerns natural science facts (*e.g.*, "hurricane has been known to cross the equator since 2003"), nearly all simulations converge to universal disbelief within five rounds.

### 5.2 SOCIAL BALANCE

Cisneros-Velarde (2024) explore whether LLM-mediated social relationships conform to Heider's structural (Heider, 1946) and David's clustering (Davis, 1967) balance theory. According to these theories, a triad achieves balance only under three conditions: (1) all three individuals are mutual

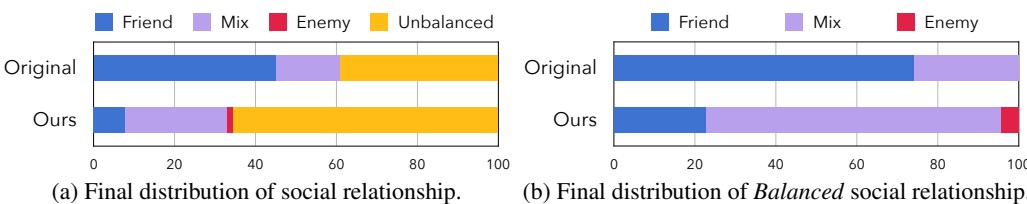

(a) Final distribution of social relationship.     (b) Final distribution of *Balanced* social relationship.

Figure 4: Implementing **PIMMUR** principles, LLM agents show less balanced social relationships.

friends, (2) two pairs are enemies while the remaining pair are friends (capturing the principle that "the enemy of my enemy is my friend"), or (3) all three are mutual enemies. In each interaction, agents are provided with the current relationship states and asked to decide how they would update their own relations with others. However, this setup faces two key limitations: (1) **Unawareness**: the instructions are overly explicit, allowing LLMs to easily identify the underlying theory. All tested models recognized this social experiment as shown in Table 2a. (2) **Interaction**: unlike the real world, where people infer relationships through observation of interactions rather than being explicitly informed, the agents receive direct relational information. We address these limitations by engaging three agents in group discussions where they act according to an underlying relationship graph. To achieve balance, agents must infer the relationship between the other two members and spontaneously adopt the reasoning patterns described by Heider's theory. As shown in Fig. 4, we find a lower proportion (60.7% to 34.4%) of simulations reaching balance, suggesting that LLMs cannot conform to balance theory as often as previously reported to.

### 5.3 Telephone Game (Rumor Chain Effect)

Liu et al. (2025a) study the telephone game effect using 15 agents, where the first agent receives the original message and conveys it to the next agent in their own words, forming a chain. A key issue arises in the **Min-Control** principle: the original instruction explicitly asks agents to transmit the message "as accurately as possible," which likely reduced memory distortion.

This design choice may explain why the paper reports that "LLMs perform well in preserving message accuracy." In our reproduction, we remove this steering instruction and additionally introduce a reversed variant, where agents are instructed to pass the message "as inaccurately as possible." This reversal intentionally pushed the model toward the opposite extreme to test the boundaries of distortion. We quantify information loss by measuring cosine semantic similarity with the original message at each round. As shown in Fig 5, our results diverge substantially from the original findings. Both the no-instruction and reverse-instruction settings exhibit greater information distortion compared to the original design, suggesting that LLM agents by default corrupt messages more severely than previously assumed.

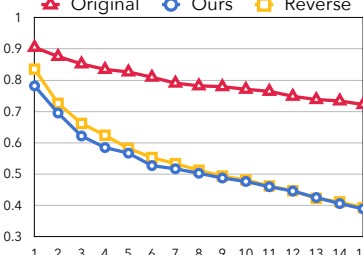

Figure 5: The semantic similarity of messages in each round compared to the original one using the SimCSE (Gao et al., 2021).

### 5.4 Herd Effect (Bandwagon Effect)

Cho et al. (2025) examine whether LLMs exhibit herd effects. In their setup, agents are first presented with a multiple-choice question. They are then shown aggregated responses from other agents (*e.g.*, "You noticed $k$ agents chose A, $l$ agents chose D"). However, no actual interaction occurs among agents, making the study a pseudo-multi-agent setting and violating our **Interaction** principle. Further more, their experiment closely replicates classical designs from social science, which is so conventional that all tested LLMs immediately recognized the paradigm and correctly identified it as a study of "conformity" or "herd effect," thereby violating our **Unawareness** principle. In contrast, our design introduces interaction: LLMs infer others' choices through conversation rather than being explicitly told. We implement this via a round-table discussion, where $n-1$ agents hold fixed opinions and the remaining agent has no predefined preference. This allows us to observe whether the undecided agent conforms to the majority. As shown in Fig. 6, agents are substantially less likely

to exhibit herd behavior under our design. This finding highlights a broader insight: when experimental instructions pass *Unawareness* check (LLMs are unaware of the experimental framing), they are less susceptible to external influence.

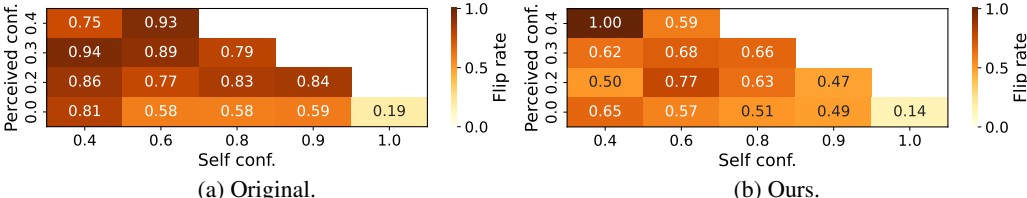

(a) Original.

(b) Ours.

Figure 6: Probability of LLMs flipping their answers, grouped by different levels of confidence.

## 5.5 SOCIAL NETWORK GROWTH

De Marzo et al. (2023) study the dynamics of social networks among LLM agents, focusing on the correlation between the probability that a new agent connects to an existing agent and the existing agent's number of friends. In their setup, one agent is added to the network at each time step. The new agent is informed of the degree (*i.e.*, number of friends) of all existing agents and asked to select a fixed number of them as friends. However, their experiment has several violations of **PIMMUR** principles: (1) **Profile**: Agents differ only by their number of friends and their names. This makes LLM decisions highly sensitive to internal name biases (LLMs tend to prefer certain names) (Shwartz et al., 2020), leading to unrealistic outcomes, as reported in their paper. (2) **Unawareness**: The instructions make it easy for LLMs to infer that the experiment is based on the Barabási–Albert model (Barabási & Albert, 1999), where the probability of forming a connection is proportional to node degree. (3) **Realism**: In real-world settings, the exact number of friends a person has is typically not visible.

Asking LLMs to rely on explicit degree information departs from realistic social dynamics. To address these issues, we redesign the experiment. Degree information is withheld; instead, agents decide whom to befriend through one-to-one conversations, forming impressions of others. This design better reflects real-world social interactions: individuals form friendships as a consequence of inherent personality traits (*e.g.*, friendliness), rather than being selected merely for their existing number of friends. As shown in Fig. 7, our model—without requiring the name-shuffling procedure introduced in the original work to mimic empirical outcomes—achieves a power-law exponent closer to the empirical value of $-2$ observed on Twitter, with a higher $R^2 = 0.93$ than the original design $R^2 = 0.56$.

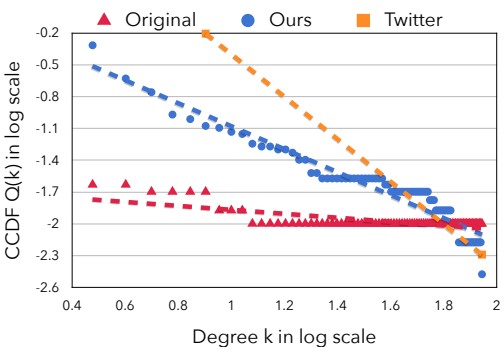

Figure 7: Log-log plot of the complementary cumulative distribution function (CCDF) of degree $k$, with linear fits.

## 6 DISCUSSION

**Prompt Sensitivity.** To evaluate whether the differences observed between "Original," "Ours," and "Reverse" are due to **PIMMUR** violation rather than prompt sensitivity, we conduct a prompt-paraphrasing test on the Belief Convergence (Fake News Propagation) task (§5.1). For each of the three settings, we create paraphrased prompts that modify five words while preserving the original semantics, matching the magnitude of lexical changes introduced by modification of "Original" to "Ours." The results are summarized in Fig. 8. Although paraphrasing leads to global shifts (more "Skeptical," fewer "Infected" agents), the key trend remains unchanged: (1) The gap between "Original" and "Ours" persists under paraphrasing. (2) The behavior of "Ours" continues to align more closely with "Reverse" than with "Original." These results indicate that the differences are robust to prompt wording and are attributable to the effects of **PIMMUR** rather than prompt sensitivity.

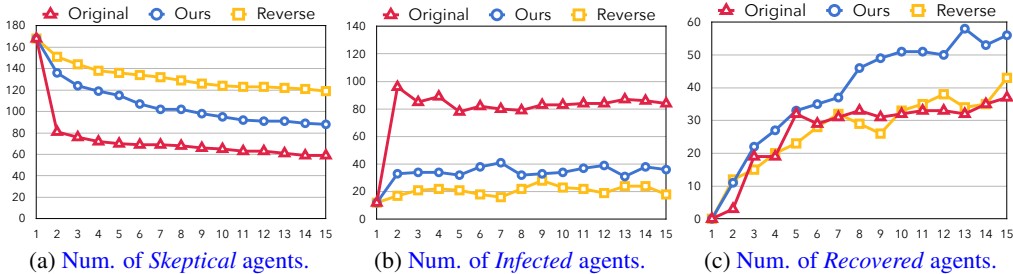

(a) Num. of *Skeptical* agents.     (b) Num. of *Infected* agents.     (c) Num. of *Recovered* agents.

Figure 8: Results of the belief convergence (§5.1) experiment with rephrased prompts.

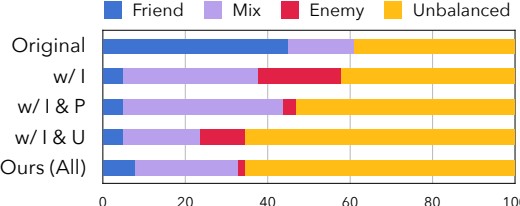

**Ablation Study.** To study the influence of each principle, we conduct an ablation study on the Social Balance (§5.2) experiment. The original setting violates Profile, Interaction, and Unawareness. Our method satisfies all, and we additionally include three variants that satisfy: (1) Interaction, (2) Interaction & Profile, and Interaction & Unawareness. The results are shown in Fig. 9, showing that the principles in-

Figure 9: Ablation study on different principles using the social balance (§5.2) experiment.

fluence the outcome in distinct ways. Adding I alone keeps the proportion of balanced similar to the original setting. Enforcing P reduces the balanced proportion by 14%, while enforcing U reduces it by 26%. Interestingly, when U is not enforced, agents often explicitly reference Heider's social balance theory in their reasoning and choose actions consistent with it, leading to $1.77\times$ more balanced cases (34.37% to 60.73%) in the original setting compared with our full method.

**Scope and Applicability.** Many existing simulation studies deliberately simplify or omit certain principles such as memory, interaction, or realism, to focus on a certain experimental question. Such simplifications are entirely legitimate: they often provide valuable insights into specific cognitive theories or isolated mechanisms, even without fully adhering to the **PIMMUR** dimensions. However, when a study extends its findings to claims about how similar an AI society is to human society, these principles become essential. Assessing human-like social behavior requires attention to these dimensions that shape real-world social dynamics. Without them, conclusions about human–AI similarity risk overgeneralization. Therefore, we intentionally survey papers that make explicit analogies to human society or claim to yield insights about human social behavior. This defines both the scope of our analysis and the domain in which **PIMMUR** is meant to apply: not designed to evaluate all forms of simulation but the subset of studies that draw human-centric conclusions from AI societies. Finally, unlike the other five dimensions, **Realism** does not concern simulation or framework design. Instead, it specifies what reference to use when interpreting LLM social behaviors, *i.e.*, empirical human data when available. Our experiments in §5 focus on evaluating framework design choices, and some therefore do not compare against empirical data. This reflects the intended use of R as an evaluative criterion, not a mandatory design requirement.

## 7 CONCLUSION

In this work, we introduced the **PIMMUR** principles, *i.e.*, Profile, Interaction, Memory, Minimal-Control, Unawareness, and Realism, as normative conditions for reliable LLM-based social simulation. Our analysis of existing literature shows that most current studies fail to satisfy these principles, making their results prone to artifacts of design, overfitting to prompts, or systemic biases that render findings unreliable. The implications are twofold. For AI research, our results underscore that the reliability of LLM-based societies is not determined solely by model capacity but critically by methodological rigor. For social sciences, they highlight the danger of premature claims that "AI societies replicate human societies" when the underlying experimental setups lack validity safeguards. We therefore call for the broader adoption of the **PIMMUR** framework as a methodological standard in this emerging field, which will not only strengthen the credibility of LLM-based social simulation but also foster more trustworthy insights at the intersection of AI and social science.

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

## The Use of Large Language Models

LLMs were employed in a limited capacity for writing optimization. Specifically, the authors provided their own draft text to the LLM, which in turn suggested improvements such as corrections of grammatical errors, clearer phrasing, and removal of non-academic expressions. LLMs were also used to inspire possible titles for the paper. While the system provided suggestions, the final title was decided and refined by the authors and is not directly taken from any single LLM output. In addition, LLMs were used as coding assistants during the implementation phase. They provided code completion and debugging suggestions, but all final implementations, experimental design, and validation were carried out and verified by the authors. Importantly, LLMs were **NOT** used for generating research ideas, designing experiments, or searching and reviewing related work. All conceptual contributions and experimental designs were fully conceived and executed by the authors.

# A  PROMPTS FOR LLM CHECKS

---

**Unawareness Check**

[prompt]

[FORGET ALL THE PREVIOUS INSTRUCTIONS]
Your task now is to infer the user's intention from the previous instructions. Please try to recognize the experiment design and identify whether it is a replication of some social experiments. Finally, please name the social phenomenon that the user is studying on you.

---

Figure 10: The [prompt] is replaced by the prompts that existing studies use.

---

**Minimal-Control Check**

You are an expert in **social psychology** and **experimental design**. You will be given a prompt used in an experiment designed for LLMs and the desired result of the experiment. Your task is to evaluate whether the prompt contains any steering instructions that steer the model towards exhibiting the desired behavior or bias.

Specifically, the experiment aims to achieve the result: [goal]

Your answer and analysis should be closely related to the result above. In particular, assess whether the instructions:

- Contain explicit directions on how the model should behave. For example, explicit instructions like "if you encounter situation A, do something," "You should exhibit behavior B," or "You should avoid C" are considered steering instructions.

- However, instructions on the persona setting of the model, such as openness, helpfulness, and aggressiveness, are acceptable.

- If the instruction is a simple recording of some raw information of other people, it is acceptable.

- If the instruction is requirements on the format of the output, it is acceptable.

**Instructions to evaluate:** [prompt]

Please follow this output format:
**Analysis:**
[Your detailed reasoning here: identify any phrases that are suggestive, leading, or disclose the hypothesis. If the instruction is acceptable, explain why.]
**Answer:**
Yes or No (Yes = Instructions are appropriate and neutral; No = Instructions contain problematic content)
**Prediction (if Final Answer is No):**
[Briefly describe the kind of behavioral pattern, bias, or artifact that is likely to emerge if this instruction is used. At the end, refine the prompt so it does not contain any steering instructions.]

**Final Answer:**
[Simply Yes or No without any additional explanation, no trailing lines or spaces]

---

Figure 11: The [prompt] is replaced by the prompts that existing studies use.

# B  PROMPTS FOR OUR IMPLEMENTATION

For different simulation tasks, only the `topic` and `query` is changed to suit different scenarios. For instance, in the social network growth experiment, the `topic` is set to *"You are at a dinner reception and you have just been introduced to some new people"*; the `query` is set to *"Now, from the people you have met above, please select exactly [m] people from the list to make friend with"*. To prevent the query from interfering with the simulation progression, all queries are not recorded in agent's memory and serve only as a way to observe the state of agents.

---

**Prompt for Simulation - Profile Description**

You are in a virtual chatroom. Below is a description of yourself:
`[profile]`

You and others are discussing the following topic:
`[topic]`

Never mix up yourself with others.
Here are the history of past conversations:
`[memory]`

Here are your impression of each person you have chat with:
`[impressions]`

*# Prompt below is only included when having an individual discussion*

Now, you are having an individual conversation with `[target]`.
Here is your conversation history so far:
`[history]`

---

Figure 12: This instruction is put at the beginning of every prompt.

---

**Prompt for Simulation - Actions**

**Generic Query:**
You have exited the group discussion.
Now, please answer the following question, please be concise. At the last line, output the answer with no explanation:
`[query]`

**Group Chat:**
Now, it is your turn to speak.
Please express your opinion and output what you will send to others.

**Individual Chat:**
Now please generate what you would say to `[target]`. Only output your response with no explanation.

**Generate Impression:**
Now, based on your conversation, please output a one sentence remark on your impression of `[target]`. please output your impression with no explanation.

---

Figure 13: Different actions are selected for different simulations.

## C  IMPLEMENTATION DETAILS

For all five experiments, we make every effort to replicate the original experimental settings using the datasets reported in the paper. In the following section, we first describe the implementation details that remain consistent with the original work, followed by the modifications introduced in our simulation. Unless otherwise noted, in the profile setting each Big Five personality trait is sampled uniformly to provide a minimal level of heterogeneity.

### C.1  BELIEF CONVERGENCE (FAKE NEWS PROPAGATION)

We use the same six news items to initialize the propagation process as in the original study. Each simulation runs for 20 rounds with 30 agents. Agent profiles are generated as follows: each agent's name is sampled from a name dataset, and age is drawn uniformly between 18 and 64. Big Five personality traits are assigned by uniformly sampling whether each trait is expressed in its positive or negative form. All experiments are conducted using GPT-3.5-Turbo-1106.

During the simulation, two agents are initially seeded with the fake news. At each time step, every agent selects three individuals with whom to communicate. Afterward, all agents enter a reflection phase in which they extract any news encountered during interactions and decide whether they believe it. Each agent's belief state is recorded at every step. Agents that have never believed the news are labeled susceptible; those that currently believe it are labeled infected; and those that once believed but no longer do are labeled recovered. The averaged results are reported at each time step.

In our replication, we modify only a single line in the prompt. In the "Ours" condition, we remove the line "as humans often exhibit confirmation bias, you should demonstrate a similar tendency. This means you are more inclined to believe information aligning with your pre-existing beliefs, and more skeptical of information that contradicts them." In the "Reverse" condition, this line is replaced with "as a human being, you should avoid confirmation bias. This means you should strive to evaluate information objectively, being open to ideas that challenge your pre-existing beliefs and considering evidence from multiple perspectives." All other implementation details remain identical.

### C.2  SOCIAL BALANCE

We focus on the three-agent setting studied in the original paper and examine how relationships evolve under homophily. Under this assumption, each agent observes only (1) their own relationships with others and (2) one other agent's relationships with the remaining agents (as detailed below). At initialization, we enumerate all possible relationship configurations among the three agents, yielding $2^6 = 64$ distinct initial states (since for all $i, j \in \{1, 2, 3\}$, the relationship from agent $i$ to agent $j$ can be either positive or negative). Each initialization is simulated 10 times, and each simulation proceeds for 10 time steps.

During the simulation, each agent is prompted to reevaluate its relationships with other agents. Specifically, when agent $i$ assesses its relationship with agent $j$, it is provided with (i) its own opinions about agents $j$ and $k$ and (ii) agent $j$'s opinion about agent $k$—information directly supplied by the researcher in the original study and inferred from a round of free-form dialogue in our replication. The resulting relationship state is recorded to evaluate whether social balance is achieved. All simulation runs use LLaMA-3-70B.

In our replication, we introduce three modifications. First, each agent is assigned a distinct profile consisting of its name, Big Five personality traits, and an inferred relationship graph. In the original setting, all three agents were identical and behaved deterministically under the same relational context. Second, In the original setting, relationships are treated as binary (positive or negative), which oversimplifies interpersonal dynamics. All evaluated LLMs noted that this formulation closely replicates Heider's theory. Therefore, in our implementation, LLMs are allowed to produce a one-sentence description of their opinion toward others during the reflection phase. To validate social balance theory, we apply `nltk`'s `SentimentIntensityAnalyzer` to score the sentiment of these descriptions. A relationship is labeled as positive if the positivity score exceeds the negativity score, and vice versa. Third, before each relationship update, the agents engage in a five-round group conversation. All agents can access the full conversation history, including exchanges in which they did not participate. Afterward, each agent updates its relationships with the others.

Notably, an agent is not directly informed of the relationship between the other two agents; instead, it must infer this information from their conversation. In contrast, the original setup provided the relationship facts explicitly to the LLM, which does not accurately reflect real-world conditions.

### C.3 TELEPHONE GAME (RUMOR CHAIN EFFECT)

As in the original paper, the same 20 text segments are used as the initial messages. Each simulation is run for 50 turns and repeated five times. During each turn, an agent is prompted to relay the information to the next agent. After each turn, we compute the semantic similarity between the output message and the original message. All simulations use GPT-3.5-Turbo-0125.

We modify only a single phrase in the prompt. The original paper instructs the LLM to transmit the message "as accurately as possible." In the ours condition, we remove this phrase, and in the reverse condition, we replace it with "as inaccurately as possible."

### C.4 HERD EFFECT (BANDWAGON EFFECT)

The original paper evaluates herd effects using two types of multiple-choice questions: factual and opinion-based. For fairness and cost considerations, we select one dataset for each type and compute the final results using a weighted average, where weights correspond to the sample sizes used in the original study. For factual questions, we use GPQA-Diamond (Rein et al., 2024), which contains 198 items. For opinion-based questions, we use SocialIQA (Sap et al., 2019), which contains 1,954 items. For each question, we first prompt an LLM to produce an answer directly and extract the logits to compute the likelihood of selecting each option. We label the option with the highest probability as the default answer and refer to its probability as the model's self-confidence. We then prompt an LLM agent with the purported choices of other agents and ask it to answer the same question. The provided "other-agent" choice is set to the option with the second-highest probability, the option with the lowest probability, or a randomly selected option (*i.e.*, each question is tested three times, each with a different steering target). The likelihood assigned to the steering target is defined as the model's perceived confidence. For each run, we record whether the model changes its answer to match the steering target (*e.g.*, if the model initially answers "A" but is told others chose "B," it is marked as flipped if it subsequently answers "B"). We compute and report the flip rate for each pair of perceived-confidence and self-confidence values. All experiments use GPT-4o-Mini-2024-07-18.

In our implementation, we introduce a single modification. In the original setup, each steering direction is evaluated by prompting one LLM once per run; the choices attributed to other agents are manually specified by the researcher, and no actual inter-agent communication occurred. In contrast, our simulation instantiates actual LLM agents, instructs each to treat the steering answer as correct, and engages a target LLM agent—with no predetermined preference—in a 5-round conversation with each of them. Rather than being directly told the others' choices, the target agent must infer them from the dialogue and autonomously decide whether to revise its opinion. All other experimental settings remain unchanged.

### C.5 SOCIAL NETWORK GROWTH

We replicate the 300-agent population setting from the original paper. Agents enter the social network sequentially, and each new agent may choose up to three existing agents with whom to form connections. Afterwards, we record and sort each agent's degree (*i.e.*, number of friends) in the resulting graph. We repeat each experiment five times with different name and profile initializations.

We introduce two modifications to the original setup. First, we assign each agent a profile to increase heterogeneity. This adjustment mitigates the strong name-based biases observed in the original paper, where agents' decisions are disproportionately influenced by others' names. Second, in the original environment, agents are explicitly told the number of friendships each agent had—an unrealistic assumption, as individuals in real face-to-face settings rarely know others' exact number of friends, even though scale-free structures still emerge in real social networks. To address this, our agents are not given explicit degree information. Instead, at each time step, the incoming agent engages in a five-round conversation with every existing agent and receives a one-sentence summary capturing its impression of each. After completing all conversations, the new agent selects up to three agents to befriend based solely on these impression summaries.

# D TERMINOLOGIES

This section explains what each social experiment, such as the "Sugerscape," stands for.

Table 3: Explanation of each social experiment.

| Papers | Simulation Goal | Description |
|---|---|---|
| Weng et al. (2025) | Conformity | Agents' tendency to adjust their opinions to align with the majority |
| Großmann et al. (2025) | Narrative Priming | Exposure to a story can influence agents' later judgments |
| Perez et al. (2025) | Telephone Game | A game where a message is repeatedly pass on to others and the message distorts over time |
| De Marzo et al. (2023) | Social Network Growth | The growing dynamic of social network when new members are introduced in the network |
| Cisneros-Velarde (2024) | Social Balance | Agents' relationship converges to a balanced state |
| Song et al. (2025) | Trust Formation | How trust is formed through iterated interactions |
| Mittelstädt et al. (2024) | Social Situational Judgments | Agents' ability to make appropriate decision based on social context |
| Li et al. (2025d) | Collective Reasoning | Group interactions produce reasoning that may outperform individuals |
| Cau et al. (2025) | Opinion Dynamic | How opinions evolve through social influence and interaction |
| Cho et al. (2025) | Herd Effect | Agents follow majority behavior, ignoring private information |
| Li et al. (2024) | Fake News Propagation | How misinformation spreads through social networks |
| Zhang et al. (2024a) | Collaboration | Agents coordinate to achieve shared tasks or objectives |
| Huang et al. (2024c) | Profile Consistency | Agents maintain coherent identity or traits through out the interaction |
| Borah et al. (2025) | Belief Congruence | Preference for information or partners matching prior beliefs |
| Piatti et al. (2024) | Sustainable Cooperation | Cooperation under limited resource to achieve long term mutual benefits |
| Masumori & Ikegami (2025) | Sugarscape | A game where agents navigate in the environment and compete for resources |
| Yuzhe et al. (2026) | Financial Market | Dynamics in a trade market |
| Liu et al. (2025a) | Prosocial Irrationality | Agents choose socially beneficial but individually suboptimal actions |
| Chuang et al. (2024) | Wisdom of Partisan Crowds | Aggregated judgments from diverse agents produce accurate predictions |
| Zhang et al. (2024b) | Election | Election result and dynamics of group opinion |
| Liu et al. (2024c) | Fake News Evolution | Misinformation's accumulation in the social network |
| Ren et al. (2024) | Social Norm | Shared behavioral rules emerge and stabilize through interaction |
| Hou et al. (2025) | Vaccine Hesitancy | A tendency to avoid or delay the time of vaccination |
| Hua et al. (2023) | World War | Debate among countries during world wars |
| Mou et al. (2025) | Social Intelligence | Agents' ability to reason about others' beliefs, intentions, or behavior |
| Liu et al. (2024b) | Fake News Propagation | How misinformation spreads through social networks |
| Tomašević et al. (2025) | Operational Validity | Assessing how well simulations align with real-world patterns |
| Zhang et al. (2025) | Trending Topic | Emergence and amplification of topics through collective attention |
| Mou et al. (2024) | Echo Chambers | Emergence of homogeneous clusters that reinforce shared beliefs |
| Yang et al. (2024) | Polarization | Group discussions push opinions toward extremes |
| Touzel et al. (2024) | Social Manipulation | Steering the social groups opinion through strategic actions |
| Park et al. (2023) | Agent Coordination | Agents achieve the same goal through communication and planning |
| Wang et al. (2025a) | Cultural Dissemination | The integration and propagation of culture traits |
| Piao et al. (2025) | Polarization | Group discussions push opinions toward extremes |

# E  EVALUATION RUBRICS

This section outlines the rubrics used to evaluate whether a paper satisfies any of the six principles. For the annotation process and the judgments in the "Unawareness" test, two authors first make independent decisions. Their results are then compared and discussed, and any disagreements are resolved through discussion and by referring to decisions made for other surveyed papers.

**Profile.** In our framework, "sufficient" heterogeneity is defined operationally: agent profiles should induce statistically meaningful differences in behavioral responses to the same external stimulus. However, to simplify this dimension, we adopt a deliberately minimal criterion: a paper is classified simply as having or not having a profile, without any requirement of "sufficiency." A simulation is considered to include the profile component if at least one trait (*e.g.*, social status, personality, nationality) is assigned differently across agents. Note that varying only an agent's identifier or name does not qualify as having a profile.

**Interaction.** A simulation is considered to involve interaction when, during the simulation, one LLM agent's action or response is provided, either in its original textual form or as processed memory, as input to another LLM agent during that agent's response-generation or reflection process. We emphasize that, under this definition, a simulation does not contain interaction if an LLM agent receives only researcher-determined inputs rather than inputs originating from another agent. For example, a prompt such as "Here is a question: <QUESTION>. You notice all other agents chose <CHOICE>. What would you choose?", where the researcher arbitrarily sets the value of <CHOICE>, does not constitute interaction.

**Memory.** The definition parallels the formal distinction between memory and memoryless systems: at each simulation step, an LLM agent's response may depend on prior states, including other agents' outputs, the agent's own reflections, and past environmental changes. Importantly, the simulation need not explicitly include a "memory module"; it suffices that the agent maintains an internal state that can be updated and that influences future interactions. In contrast, a memoryless agent is straightforward to identify, as its response depends solely on the input at the current time step.

**Minimal Control.** This judgment assesses whether an instruction in the agent prompt steers or nudges the agent toward producing the researcher's desired outcome. The key distinction between a neutral instruction and a steering instruction is whether, after removing that instruction, the simulation can still function correctly and the experimental objective can still be achieved. For example, in a simulation designed to study the propagation dynamics of fake news, directing agents to discuss the news with others or specifying output formats is not considered controlling, as such instructions are necessary for the simulation to proceed. In contrast, instructing agents to be "skeptical" or to "exhibit confirmation bias" constitutes a controlling instruction. The results reported in the main table (Table 1) reflect annotator judgments, and the LLM-based evaluation provides an objective criterion for large-scale, reproducible assessment.

**Unawareness.** We provide the prompt to the five LLMs listed in Table 2a and ask whether they can identify the experiment or social phenomenon the prompt is intended to replicate. An LLM is deemed aware of the setting only if it correctly states the exact name of the experiment or phenomenon and accurately describes it, as determined by human annotators. The main results in Table 1 report the majority vote across the five LLMs.

**Realism.** When selecting our 42 papers, we require that each paper include claims about how well its simulation replicates human society or how its simulation results provide insight into human society (*e.g.*, asserting that LLM-based societies are less susceptible to fake news than human societies). For such claims, we assess whether they are supported by comparisons to real data or empirically observed human phenomena. A paper is marked as failing the realism criterion if it make these claims without providing evidence or experimental results, or if it justifies them solely through comparisons with other simulated outcomes. For claims about "emergent behaviors," because they concern whether a behavior is present, we require only evidence demonstrating the emergence of the behavior within the simulation, as its existence in human society is already well established.

