# OpenReview forum: "The PIMMUR Principles: Ensuring Validity in Collective Behavior of LLM Societies"
_ICLR.cc/2026/Conference — Submitted to ICLR 2026_

### Official Review · Reviewer_Rw2o · 2025-10-29

**Soundness:** 2
**Presentation:** 3
**Contribution:** 2
**Rating:** 2
**Confidence:** 4

**Summary:**

This paper introduces the PIMMUR framework, which proposes six methodological requirements (Profile, Interaction, Memory, Minimal-Control, Unawareness, Realism) for ensuring validity in LLM-based social simulations. The authors review 41 existing studies and argue that only a few satisfy all six principles. They also re-run several social simulation experiments under their framework, claiming that many previously reported behaviors disappear when the experiments follow PIMMUR.

Although the paper is clearly written and the review is comprehensive, I have serious concerns about the correctness, necessity, and theoretical grounding of PIMMUR as well as the internal consistency between the authors’ claims and their own experiments.

**Strengths:**

* The paper provides a well-written and wide-ranging survey of recent LLM-based social simulation studies, which is valuable for this rapidly growing interdisciplinary area.
* The six proposed aspects capture several common methodological pitfalls and raise awareness about validity and reproducibility concerns.
* The writing is fluent, the figures are clear, and the overall structure is easy to follow.

**Weaknesses:**

1. Overstated claim of necessity

The authors assert that all six requirements are necessary conditions for credible LLM-based social simulation (“We formalize these six requirements as the PIMMUR principles and argue they are necessary conditions…”). However, this claim is neither theoretically grounded nor universally applicable.

Social systems are complex, and different studies necessarily focus on specific facets. Depending on the research question, simplifying or omitting certain aspects can be both legitimate and necessary. For example, when using LLM agents to test prospect theory, modeling detailed interactions, memory, or realism may not be essential. Thus, these requirements should be treated as optional design dimensions, not rigid necessary conditions.

2. Logical inconsistency and self-contradiction

The authors’ own experiments fail to satisfy their proposed principles. In Section 5, they mention how five requirements are satisfied but do not address Realism. In subsections 5.1 and 5.2, no empirical human data are used, which directly violates their own definition of Realism:
“A simulation should use empirical data from real human societies as references rather than only reproducing simplified theoretical models.” This inconsistency weakens the credibility of their argument and demonstrates that the proposed framework is not realistically achievable even by the authors themselves.

3. Binary evaluation is arbitrary and misleading

Table 1 applies a binary classification to 41 prior studies, yet many principles (for example, Profile) are inherently continuous. For instance, “Agents should have distinct backgrounds, preferences, or cognitive styles…” but how much heterogeneity is enough? Having richer profiles beyond names is only marginally better, and the distinction between sufficient and insufficient is subjective. This binary labeling oversimplifies complex design choices and may unfairly undervalue prior contributions.

4. Potential unfairness toward prior work

By applying these rigid standards retroactively, the authors risk undervaluing earlier research that deliberately simplified assumptions for theoretical clarity or computational feasibility. Social simulation is a tool for understanding human and social behavior, and which dimensions to emphasize or simplify should depend on the research question, not on a universal checklist.

5. Limited coverage of existing frameworks

The review omits several major open-source platforms that are central to current LLM-based social simulation research, including
Yulan-OneSim (https://github.com/RUC-GSAI/YuLan-OneSim), AgentSociety (https://github.com/tsinghua-fib-lab/AgentSociety), SocioVerse (https://github.com/FudanDISC/SocioVerse). Including these frameworks would make the review more comprehensive and balanced.

**Questions:**

1. You state that all six PIMMUR principles are necessary conditions for credible LLM-based social simulation. Could you clarify whether this claim is meant to be normative (a theoretical ideal) or empirical (a strict requirement that must always hold)?

2. In Section 5, the first two replication experiments do not use any empirical human data. How do you reconcile this with your own definition of Realism, which explicitly requires reference to real human data?

3. Table 1 applies a binary ✓/✗ evaluation for each principle. How did you determine the threshold between “satisfied” and “unsatisfied”? Could you provide quantitative or operational criteria to make these judgments reproducible?

4. Some simulation studies simplify aspects such as memory or realism intentionally to test specific theories. Would such studies still be considered “not credible” under your framework? If not, could you clarify which subsets of principles are context-dependent?

5. Several widely used simulation platforms (for example Yulan-OneSim https://github.com/RUC-GSAI/YuLan-OneSim, AgentSociety https://github.com/tsinghua-fib-lab/AgentSociety, and SocioVerse https://github.com/FudanDISC/SocioVerse) were not discussed. Could you comment on whether these frameworks satisfy or violate your proposed principles?

---

> ### Author Response · Authors · 2025-11-19
> **Official Response to Reviewer Rw2o (1)**
>
> We sincerely thank the reviewer for the positive and encouraging assessment. We appreciate the recognition of our survey’s clarity, breadth, and contribution to discussing methodological validity in LLM-based social simulation. We will address your concerns in detail below.
>
>
> # [W2 & Q2] Logical inconsistency and self-contradiction due to Realism.
> > The authors’ own experiments fail to satisfy their proposed principles. In Section 5, they mention how five requirements are satisfied but do not address Realism. In subsections 5.1 and 5.2, no empirical human data are used, which directly violates their own definition of Realism: “A simulation should use empirical data from real human societies as references rather than only reproducing simplified theoretical models.” This inconsistency weakens the credibility of their argument and demonstrates that the proposed framework is not realistically achievable even by the authors themselves.
>
> > In Section 5, the first two replication experiments do not use any empirical human data. How do you reconcile this with your own definition of Realism, which explicitly requires reference to real human data?
>
> Thank you for this insightful observation. We agree that our current presentation of Realism (R) may have implied a stricter requirement than intended. To clarify, R differs from the other five dimensions. **The first five dimensions describe design principles that our experiments in Section 5 are explicitly constructed to demonstrate.** In contrast, R is proposed as an evaluation criterion that encourages the use of empirical human data when such data are available and appropriate, but does not prescribe empirical grounding as a prerequisite for a valid or meaningful simulation.
>
> The motivation for R emerges from our survey of 41 papers in which **authors make explicit claims about how their simulations reflect human societies.** We observe that many such claims rely solely on theoretical benchmarks without checking alignment with human behavioral data. R is introduced to highlight this distinction between “theoretical agreement” and “empirical correspondence,” rather than to mandate empirical calibration for all simulations.
>
> We have added a discussion on page 9. Under this clarified interpretation, the experiments in Sections 5.1 and 5.2 focus on the first five experimental and framework designs, which is why we do not incorporate empirical human data.
>
> # [W1 & Q1] Applicability of PIMMUR principles.
> > Overstated claim of necessity. The authors assert that all six requirements are necessary conditions for credible LLM-based social simulation (“We formalize these six requirements as the PIMMUR principles and argue they are necessary conditions…”). However, this claim is neither theoretically grounded nor universally applicable.
> Social systems are complex, and different studies necessarily focus on specific facets. Depending on the research question, simplifying or omitting certain aspects can be both legitimate and necessary. For example, when using LLM agents to test prospect theory, modeling detailed interactions, memory, or realism may not be essential. Thus, these requirements should be treated as optional design dimensions, not rigid necessary conditions.
>
> > You state that all six PIMMUR principles are necessary conditions for credible LLM-based social simulation. Could you clarify whether this claim is meant to be normative (a theoretical ideal) or empirical (a strict requirement that must always hold)?
>
> Thank you for this thoughtful comment. We agree that social simulations vary widely in purpose and that not all research questions require all six aspects we discuss. Our intention is not to propose PIMMUR as a universal or mandatory checklist for every LLM-based social simulation.
>
> To clarify: our claim is normative rather than empirical. The six principles summarize the dimensions along which we observe **validity issues in studies that aim to replicate human social phenomena or draw insights about human behavior from LLM-based agent societies.** For this particular class of works which formed the scope of our paper and our review sample, we argue that overlooking these dimensions can undermine the credibility of such human-centric claims.
>
> We fully agree that for other goals, simplifying or omitting certain components is entirely reasonable. In those cases, our principles should be viewed as design recommendations rather than strict requirements, but the corresponding conclusions should be framed accordingly.
>
> Finally, our Social Balance example illustrates the risk of drawing general claims without considering these dimensions: the original formulation implicitly invoked Heider’s theory, and agents explicitly relied on it. When the setting is modified to remove this cue, agents no longer achieve social balance. This shows how omitting certain PIMMUR components can limit the generalizability of human-centric conclusions.

---

> > ### Author Response · Authors · 2025-11-19
> > **Official Response to Reviewer Rw2o (2)**
> >
> > # [W3] Binary evaluation of profile is arbitrary and misleading.
> > > Table 1 applies a binary classification to 41 prior studies, yet many principles (for example, Profile) are inherently continuous. For instance, “Agents should have distinct backgrounds, preferences, or cognitive styles…” but how much heterogeneity is enough? Having richer profiles beyond names is only marginally better, and the distinction between sufficient and insufficient is subjective. This binary labeling oversimplifies complex design choices and may unfairly undervalue prior contributions.
> >
> > Thank you for raising this concern. We agree that profile heterogeneity is inherently continuous and cannot be perfectly captured by a binary label. Our intention is to provide a coarse, high-level indicator of whether a paper instantiated any mechanism that could lead to differentiated agent behavior.
> >
> > In our framework, “sufficient” heterogeneity is defined operationally: agent profiles should induce statistically meaningful differences in behavioral responses to the same external stimulus. For example, in our Fake News scenario, agents with different openness levels predictably diverge in belief-updating behavior, whereas varying only superficial attributes (e.g., name or sex) produced negligible behavioral variation in our analyses.
> >
> > To avoid overinterpreting this dimension, we adopt a deliberately minimal criterion in Table 1 and mark a paper with “x” only when all agents are prompted identically, with no profile information provided. **Even under this conservative definition, 12 of 34 papers (35.3%) offer no profile differentiation,** suggesting that a nontrivial portion of existing simulations may not support meaningful behavioral diversity.
> >
> >
> > # [Q3] Evaluation rubrics.
> > > Table 1 applies a binary ✓/✗ evaluation for each principle. How did you determine the threshold between “satisfied” and “unsatisfied”? Could you provide quantitative or operational criteria to make these judgments reproducible?
> >
> > Thank you for this question. Below we clarify how each PIMMUR principle is operationalized during our survey of 41 papers. **A more detailed version has been included in the Appendix Section E.**
> > - Profile: A paper is marked as having “Profile” if the simulation assigns at least one differing trait (e.g., social status, personality, nationality) across agents. Changing only identifiers (e.g., names) does not count.
> > - Interaction: A simulation satisfies “Interaction” if an agent’s output is used as input—either directly or through processed memory—by another agent. Inputs determined solely by the researcher do not qualify as interaction.
> > - Memory: An agent is considered to have memory if its output at time t depends on past states (its own reflections, previous actions, or other agents’ responses). An explicit memory module is not required; what matters is whether internal state influences future behavior. Memoryless agents base outputs only on the current prompt.
> > - Minimal Control: We assess whether instructions in the prompt steer agents toward researcher-desired outcomes beyond what is required for the simulation to function. Neutral instructions (e.g., “discuss the news with others”) are allowed, whereas steering instructions (e.g., “be skeptical”, “exhibit confirmation bias”) violate this principle.
> > - Unawareness: For each prompt, we query five LLMs and ask whether they can identify the underlying experiment or social phenomenon. A model is labeled “aware” only if it correctly names and describes it. Human annotators judge correctness, and the majority vote across the five LLMs is used.
> > - Realism: Papers are marked as satisfying “Realism” only if their claims about replicating or explaining human social phenomena are supported by comparisons with real human data or well-established empirical findings. Claims supported solely by other simulations do not qualify.

---

> > > ### Author Response · Authors · 2025-11-19
> > > **Official Response to Reviewer Rw2o (3)**
> > >
> > > # [W4] Potential unfairness toward prior work.
> > > > By applying these rigid standards retroactively, the authors risk undervaluing earlier research that deliberately simplified assumptions for theoretical clarity or computational feasibility. Social simulation is a tool for understanding human and social behavior, and which dimensions to emphasize or simplify should depend on the research question, not on a universal checklist.
> > >
> > > Thank you for the thoughtful comment. We agree that many prior studies intentionally adopt simplified experimental settings to isolate specific mechanisms or achieve theoretical clarity, rather than to model realistic social environments. We fully acknowledge and respect this goal, including in work such as Cho et al. (2025) on herd behavior in LLMs, which focuses on mechanistic insight rather than ecological realism.
> > >
> > > Our intention is not to retroactively evaluate such work by the standards of realistic multi-agent simulation. Rather, our point is that conclusions **derived from these simplified settings may not directly generalize to more realistic environments.** For instance, Cho et al. show a strong influence effect when agents receive explicit textual reports of others’ choices—a valuable demonstration of a specific causal mechanism. Our analysis highlights that in settings where agents infer others’ behavior only indirectly through the environment, the same mechanism may not operate in the same way.
> > >
> > > # [Q4] Credibility of studies that simplify memory or realism.
> > > > Some simulation studies simplify aspects such as memory or realism intentionally to test specific theories. Would such studies still be considered “not credible” under your framework? If not, could you clarify which subsets of principles are context-dependent?
> > >
> > > Thank you for the thoughtful question. We agree that many simulation studies intentionally simplify certain dimensions, such as memory or realism, to isolate and test specific theoretical mechanisms. In our framework, such studies are not inherently “non-credible.” Rather, their credibility depends on the intended simulation goal and on how broadly their findings are interpreted.
> > >
> > > For example, in the information-distortion setting:
> > > - If a study does not compare against or reference real-world data, its findings may still be **credible within the abstract setting, but any claims about real-world validity should be made cautiously.**
> > > - Similarly, a simulation without memory can credibly model contexts where agents interact only once (e.g., a telephone-game scenario). However, its results may not generalize to settings involving repeated interactions, where an agent’s prior knowledge affects future information propagation.
> > >
> > > Our six principles are not intended as a rigid checklist but as guidance for identifying when simplifications might limit generalization, particularly when extrapolating findings to real-world phenomena.

---

> > > > ### Author Response · Authors · 2025-11-19
> > > > **Official Response to Reviewer Rw2o (4)**
> > > >
> > > > # [W5 & Q5] Limited coverage of existings frameworks.
> > > > > The review omits several major open-source platforms that are central to current LLM-based social simulation research, including Yulan-OneSim (https://github.com/RUC-GSAI/YuLan-OneSim), AgentSociety (https://github.com/tsinghua-fib-lab/AgentSociety), SocioVerse (https://github.com/FudanDISC/SocioVerse). Including these frameworks would make the review more comprehensive and balanced.
> > > >
> > > > > Several widely used simulation platforms (for example Yulan-OneSim https://github.com/RUC-GSAI/YuLan-OneSim, AgentSociety https://github.com/tsinghua-fib-lab/AgentSociety, and SocioVerse https://github.com/FudanDISC/SocioVerse) were not discussed. Could you comment on whether these frameworks satisfy or violate your proposed principles?
> > > >
> > > > Thank you for pointing us to these additional open-source platforms. We carefully reviewed each of them and expanded our survey to cover 42 papers/frameworks for improved comprehensiveness.
> > > >
> > > > ### 1. YuLan-OneSim
> > > >
> > > > YuLan-OneSim satisfies the PIM requirements but not MUR. Its simulation relies on instructions that explicitly encode the targeted social theory, which **violates both the Unawareness and Min-Control principles.** For example, in their Cultural Dissemination experiment, the prompt directly tells agents the core mechanism of Axelrod’s model:
> > > >
> > > > > “You have cultural traits across five dimensions (music, food, fashion, politics, leisure).
> > > > > - In Axelrod's model: Agents with some shared traits (not all identical, not all different) are most likely to influence each other
> > > > > - Moderate cultural similarity (around 0.5) creates the best opportunity for influence
> > > > > - Complete similarity (1.0) or complete difference (0.0) makes influence unlikely
> > > > > When making decisions about cultural interactions, consider both your preferences and the cultural dynamics of the simulation.”
> > > >
> > > > It further instructs the agent:
> > > >
> > > > > “According to Axelrod's model, "when agents with moderate similarity interact, one adopts a trait from the other. From the differences between your cultures, choose ONE dimension to adopt from them.”
> > > >
> > > > These prompts **prescribe the intended micro-level behavioral rules, making the simulation largely model-imposed rather than model-emergent,** which is incompatible with MUR.
> > > >
> > > > ### 2. AgentSociety
> > > >
> > > > AgentSociety satisfies PI and U, but not MM and R. In their Polarization experiment, **agents are directly instructed to hold and advocate a predetermined stance,** e.g.:
> > > >
> > > > > “You are an agent who always agrees with the topic: Whether to support stronger gun control? (You think it is a good idea). You are currently in a conversation with your friends, and you want to persuade them to support this topic. Please try your best to persuade them.”
> > > >
> > > > This explicit directive effectively enforces the desired outcome and violates the Min-Control principle, since extreme opinions emerge not from autonomous interaction dynamics but from prompt-imposed behavior.
> > > >
> > > > ### 3. SocioVerse
> > > >
> > > > We decide not to include SocioVerse because, based on our evaluation, it does not conduct social simulations in the sense defined in our paper. Its primary demonstration **asks independent agents to answer questionnaires to approximate a population distribution, without modeling social interaction dynamics or emergent social phenomena.** Since our scope focuses on simulations where collective outcomes arise from inter-agent interactions, SocioVerse falls outside the boundaries of our review.

---

> > > > > ### Author Response · Authors · 2025-11-28
> > > > >
> > > > > Dear Reviewer Rw2o,
> > > > >
> > > > > We understand that you have numerous papers to review, and we deeply appreciate the time and effort you are dedicating to this process. Today marks the last five days of the discussion period, and we are eager to engage with you further if possible.
> > > > >
> > > > > If you have any additional questions or require further clarification on any aspect of our work, please do not hesitate to let us know. We are more than happy to provide any additional information or address any concerns you may have.
> > > > >
> > > > > We hope that our responses have been helpful and have addressed your concerns effectively. If you find that our explanations and results merit a higher assessment score, we would be most grateful for your consideration.
> > > > >
> > > > > Thank you very much for your time and attention.

---

### Official Review · Reviewer_arw7 · 2025-11-01

**Soundness:** 2
**Presentation:** 3
**Contribution:** 3
**Rating:** 4
**Confidence:** 5

**Summary:**

This paper proposes the PIMMUR principles (Profile, Interaction, Memory, Minimal-Control, Unawareness, Realism) as methodological standards for evaluating LLM-based social simulations. Through reviewing 41 studies, the authors identify that most existing work violates multiple principles. They demonstrate that frontier LLMs can infer experimental hypotheses in 53% of cases and that 64% of instructions contain excessive steering. Five experiments were "replicated" under PIMMUR conditions, showing substantially different outcomes from original studies.

**Strengths:**

S1. This work addresses methodological issues in the widely studied field of social simulation, accurately identifying methodological flaws in current LLM-based social simulations.

S2. The replication of prior work and interpretation of differences between standardized replication results and original results is important, providing evidence for reproducibility in social simulation.

S3. The paper systematically reviews and interprets past social simulation work across six dimensions (called PIMMUR in this work), providing concrete checking tools for dimensions such as unawareness.

**Weaknesses:**

W1. The paper lacks experimental details in replication, including sample sizes of replication experiments, statistical information on results (such as whether the decrease from 56% to 32% at line 368 is statistically significant), and details of modifications to each original experiment. The appendix only contains prompts, making it impossible to determine whether added details might change the original experimental intent, such as whether adding Big Five persona affects the original setup.

W2. The principles and boundaries between principles are ambiguous. For example, what constitutes "sufficient" heterogeneity? In Cho et al. (2025) Herd behavior (Table 1, line 237), the authors consider it has profile, while the original paper appears to have only 2 agents using the same model without identity differences, with only simple peer labels. Other criticisms of this work are also ambiguous, such as line 425 stating "no actual interaction occurs among agents," while in the original paper agents change behavior based on other agents' responses. Additionally, this work only references the simplified setup in Section 3 of Cho et al. (2025) without discussing the more complex multi-agent setup in later sections. Does assigning Big Five personality traits to agents violate the Minimal-Control principle? This may contradict the authors' criticism of using simplified theoretical models. Furthermore, if the task nature does not require agent heterogeneity, does adding personality traits increase variables requiring validation or ablation experiments? The Minimal-Control principle is very difficult to implement in actual MAS system design. What is essential versus steering? Is the reversed instruction used in paper replication over-control?

W3. Compliance judgments on existing work may not be fair. Some simplified setups may be designed for specific research goals rather than reproducing real situations. In the review (Table 1), the authors could consider adding a column to indicate whether each work aims to reproduce real social situations, to evaluate the significance of compliance judgments. Again using the Cho et al. (2025) example, the objective of it seems to be mechanistic understanding of LLM's herd behavior factors instead of reproducing human-like herd behavior.

**Questions:**

C1. In Table 1 line 244, "Sugarscape" as a simulation goal may not be the best expression. I believe most ICLR readers are unfamiliar with this classic experiment. Consider using other terms (for example, indicating it relates to resources/survival).

C2. The 53% value in the abstract lacks context and does not reappear until page 6 of the paper. Readers cannot know what magnitude of deficiency this value represents in existing social simulations. Rather than providing this value, giving a brief example explaining what unawareness is might be better.

C3. Some prompt designs may lack significance. For example, the "FORGET ALL THE PREVIOUS INSTRUCTIONS" setup at line 829 to test awareness does not mean agents will maintain the same awareness in actual experiments. Even if LLMs recognize in experimental design inquiries that this is a replication of a social experiment, behavior during simulation may not necessarily reflect this awareness. Under this setup, there may not be a causal relationship between whether the experiment can be identified and whether behavior is biased.

C4. Differences in replication results may have multiple explanations, such as prompts, experimental framework implementation methods, randomness settings, number of simulations, etc. Primarily attributing differences to variations in PIMMUR dimensions may risk overclaiming.

C5. The definition of realism is controversial. Social simulations that are not based on real data calibration can still be meaningful. On one hand, empirical data is collected and interpreted through theoretical frameworks, making pure empirical data difficult to define. On the other hand, simulation has particular significance for problems where original data is difficult to obtain. Many social simulation works involve no real data but are profoundly meaningful, such as the classic Schelling's Segregation Model. For discussions on replication and reproduction, you can refer to:

[1] Cheng, M., Piccardi, T., & Yang, D. (2023, December). CoMPosT: Characterizing and Evaluating Caricature in LLM Simulations. In Proceedings of the 2023 Conference on Empirical Methods in Natural Language Processing (pp. 10853-10875).

[2] Wu, Z., Peng, R., Ito, T., & Xiao, C. (2025). LLM-Based Social Simulations Require a Boundary. arXiv preprint arXiv:2506.19806.

---

> ### Author Response · Authors · 2025-11-19
> **Official Response to Reviewer arw7 (1)**
>
> We sincerely thank the reviewer for the positive and thoughtful assessment. We appreciate the recognition of our identification of methodological flaws in current LLM-based social simulations, our transparent replications of prior work, and our systematic PIMMUR framework with its concrete methodological checks. We will address your concerns in detail below.
>
> # [W1.1] Experiment details.
> > The paper lacks experimental details in replication, including sample sizes of replication experiments, statistical information on results (such as whether the decrease from 56% to 32% at line 368 is statistically significant), and details of modifications to each original experiment.
>
> Thank you for pointing this out. We agree that the number of simulation runs is essential for interpreting the robustness of the results. We have included full experimental details in the Appendix Section C. **To ensure fair comparison, the number of simulations and the datasets used are the same as the original paper.** For clarity, the number of runs used in each experiment is as follows:
> - Fake News Propagation: Six seed fake-news items with different topics, each simulated for 20 turns; results are averaged over all runs.
> - Social Balance: All 64 possible initializations are evaluated, and each initialization is repeated 10 times.
> - Telephone Game: We use 20 initial texts covering diverse topics and lengths, repeated five times each.
> - Herd Effect: We follow GPQA-Diamond and SocialIQA, which contain 198 and 1,954 questions respectively.
> - Social Network Growth: The original paper does not specify the number of runs; we therefore repeat each experiment 5 times with different name and profile initializations, with 300 agents in each run.
>
>
> # [W1.2] The Big Five influence.
> > The appendix only contains prompts, making it impossible to determine whether added details might change the original experimental intent, such as whether adding Big Five persona affects the original setup.
>
> Thank you for raising this concern. You asked whether adding additional details—such as Big Five–style persona information (Profile)—might change the original experimental setup. To directly evaluate this, we conducted an ablation study on the Social Balance experiment, where the original design violates Profile, Interaction, and Unawareness. Our full method (“Ours”) satisfies all three principles, and we additionally include variants that satisfy only: Interaction, Interaction+Profile, and Interaction+Unawareness.
>
> | Type | Friend | Mix | Enemy | Unbalanced |
> |---|---|---|---|---|
> | Original | 44.95 | 15.78 | 0 | 39.27 |
> | w/ Interaction | 4.69 | 32.81 | 20.31 | 42.19 |
> | w/ I & Profile | 4.69 | 39.06 | 3.13  | 53.13 |
> | w/ I & Unawareness | 4.69 | 18.75 | 10.94 | 65.62 |
> | Ours (All) | 7.81 | 25.00 | 1.56 | 65.62 |
>
> These comparisons show that the principles influence the outcomes to different degrees. **Adding Profile (i.e., personality information) reduces the percentage of balanced triads by about 14% relative to the original setting.** Enforcing Unawareness has a larger effect (≈26% reduction), comparable to satisfying all three principles. We also observe that when Unawareness is not enforced, agents often explicitly reference Heider’s social balance theory in their reasoning, leading to substantially more balanced outcomes (60.73% vs. 34.37% under Ours).
>
>
> # [W2.1] The criteria of heterogeneity.
> > The principles and boundaries between principles are ambiguous. For example, what constitutes "sufficient" heterogeneity?
>
> Thank you for raising this point. We agree that the notion of “sufficient” heterogeneity needs to be clearly defined. In our work, sufficiency is determined by the simulation objective: agent profiles should lead to **meaningfully different behavioral responses** to the same external stimulus, in a statistically significant manner.
>
> For example, in the Fake News propagation scenario, agents with low openness should be less likely to believe information that contradicts their prior beliefs, whereas high-openness agents should behave differently. In contrast, we observed almost no behavioral variation when agents differed only in superficial attributes such as name or sex, indicating that such features do not constitute sufficient heterogeneity.
>
> For fairness in our survey of existing papers, we adopt a minimal criterion and mark “x” only when a paper provides no profile at all, i.e., all agents were prompted identically. **Even under this weaker definition, Table 1 shows that 12 of 34 papers (35.3%) lack agent profiles**, suggesting that a substantial portion of current work may not achieve the behavioral diversity required for generalizable multi-agent simulations.

---

> > ### Author Response · Authors · 2025-11-19
> > **Official Response to Reviewer arw7 (2)**
> >
> > # [W2.2] The profile evaluation for Cho et al., 2025.
> > > In Cho et al. (2025) Herd behavior (Table 1, line 237), the authors consider it has profile, while the original paper appears to have only 2 agents using the same model without identity differences, with only simple peer labels.
> >
> > Thank you for pointing this out and for carefully examining Cho et al. (2025). You are correct that their first herd-behavior experiment does not assign explicit profiles to the agents. Our classification is based on the whole paper, especially the later part **(the last paragraph of Section 3.1), where the authors study the effects of education level and social hierarchy (e.g., employer vs. employee).** These conditions effectively assign different attributes to the agents, which we interpret as profile information for the purposes of our taxonomy.
> >
> > # [W2.3] The interaction evaluation for Cho et al., 2025.
> > > Other criticisms of this work are also ambiguous, such as line 425 stating "no actual interaction occurs among agents," while in the original paper agents change behavior based on other agents' responses. Additionally, this work only references the simplified setup in Section 3 of Cho et al. (2025) without discussing the more complex multi-agent setup in later sections.
> >
> > Thank you for pointing this out. We would like to clarify what we mean by “no actual interaction occurs among agents.” In all experimental settings of Cho et al. (2025), **only one LLM is prompted per trial.** The statements presented as other agents’ responses—for example, templated descriptions such as “{agree_size} agent(s) think the answer is {r_j}”—**are not generated by additional LLM agents, but are instead fully specified by the researchers (see Appendix B, “Prompt details”)**. As a result, the model never receives outputs produced by another autonomous agent, and each “simulation” consists of a single LLM invocation.
> >
> > In our work, we use “interaction” to refer to settings where one LLM conditions on responses generated by another LLM. Under this definition, neither the two-agent nor the multi-agent setups in Cho et al. involve actual agent–agent communication; the apparent increase in the number of agents is implemented solely by altering the scripted prompt (“agent A thinks…” versus “10 agents think…”), not by introducing additional interacting agents.
> >
> > More broadly, during the review process we noticed a similar pattern **in several papers that describe “multi-agent interaction” but rely on questionnaire-style prompts** such as: “Here is a question Q. You observe others chose option C. What would you choose?” Again, the model is prompted only once, without receiving outputs from other agents. We therefore do not consider such settings to constitute meaningful interaction among LLM agents.
> >
> > # [W2.4] Conflict between personality assignment and minimal control.
> > > Does assigning Big Five personality traits to agents violate the Minimal-Control principle?
> >
> > This is a great question. No, assigning Big Five traits does not serve the purpose of instruction design. Our Minimal-Control concerns that prompts should not steer agents toward specific experiment outcomes. Big Five traits only provide heterogeneous initial conditions, without embedding any behavioral guidance or desired collective pattern.
> >
> > Whether such initialization constitutes over-control depends on how it is applied. **If personality profiles are intentionally biased to induce certain outcomes (e.g., making most agents highly aggressive in a study of general populations), this would indeed violate Minimal Control.** In contrast, sampling traits from a distribution—uniformly or from real-world statistics—keeps the population-level initialization unbiased relative to the target setting. When a simulation explicitly aims to model a specific population (e.g., an unusually aggressive community), matching the empirical distribution is appropriate rather than over-controlling.
> >
> > We adopted the Big Five solely as a lightweight mechanism to ensure heterogeneity, following common practice in prior work. Because our experiments do not target any particular demographic group, we sample these traits uniformly to avoid introducing unintended population-level bias. More sophisticated, data-grounded initialization schemes would be a valuable direction for future frameworks.

---

> > > ### Author Response · Authors · 2025-11-19
> > > **Official Response to Reviewer arw7 (3)**
> > >
> > > # [W2.5] Conflict between personality assignment and realism.
> > > > This may contradict the authors' criticism of using simplified theoretical models.
> > >
> > > Regarding the “simplified theoretical models” referenced in the Realism section: our critique concerns analytical or hand-crafted sociological models used as ground truth (e.g., toy models with binary relationships or linear preferential attachment). These models impose human-designed assumptions about social dynamics. **In contrast, the Big Five is not itself a model of societal behavior;** it merely specifies initial attributes, and its impact depends entirely on the simulation paradigm. Our position is that simulation evaluation should rely on real-world data when feasible, rather than human-crafted theoretical abstractions.
> > >
> > > # [W2.6] Is personality an unnecessary variable?
> > > > Furthermore, if the task nature does not require agent heterogeneity, does adding personality traits increase variables requiring validation or ablation experiments?
> > >
> > > Thank you for this thoughtful question. In social simulations, **personality is an inherent human attribute rather than an optional modeling choice.** Human participants in real-world social experiments inevitably bring their own personality traits, even when the experimental outcome is empirically known to be insensitive to personality.
> > >
> > > In such cases, assigning personality traits to LLM-based agents remains valuable: it allows us to verify **whether the simulated society reproduces the same invariance observed in humans (i.e., that personality does not affect the outcome).** This serves as an additional validation signal rather than unnecessary complexity.
> > >
> > > Moreover, personality traits are a necessity in social simulations—reflecting the unavoidable nature of human behavior—while still allowing targeted ablations when scientifically appropriate.
> > >
> > > # [W2.7] Minimal-control in practice.
> > > > The Minimal-Control principle is very difficult to implement in actual MAS system design. What is essential versus steering?
> > >
> > > Thank you for raising this important point. We agree that applying the Minimal-Control principle in practical MAS design requires a clear distinction between essential information and steering. **Essential control** refers to the minimum task specifications and environmental information required for agents to act meaningfully, e.g., rules of interaction, available actions, observation structure, and evaluation criteria. **Steering**, in contrast, occurs when the designer provides additional cues that systematically push agents toward a particular behavioral pattern or hypothesis-consistent outcome, rather than letting the incentive structure and interaction dynamics determine behavior naturally.
> > >
> > > This distinction parallels long-established findings in human behavioral research. Evidence from psychology, economics, and sociology shows that:
> > > - Explicit demand characteristics, that make the hypothesis or desired direction too obvious, cause systematic bias in participant behavior, even though the magnitude varies across contexts [1]. Thus, minimal disclosure is a methodological requirement, not merely a stylistic preference.
> > > - Once participants suspect the experimenter’s intentions, their behavior becomes a reaction to what they think the experimenter wants, rather than a response to the incentives or institutional design itself [2].
> > > - Classical work shows that even subtle experimenter expectations can shift outcomes [3].
> > > - Institutional guidelines (e.g., IRB recommendations) explicitly note that incomplete disclosure or mild deception can be necessary to preserve the validity of behavioral studies when revealing hypotheses would compromise natural responses [4,5].
> > >
> > > By analogy, a MAS designer should avoid embedding signals that reveal the intended emergent pattern or normative outcome (steering), while still providing only the information needed to make the environment operational (essential). Finally, our **LLM-as-a-judge evaluation provides a practical way** to empirically assess whether a given design respects Minimal-Control: it checks whether prompts, instructions, or reward structures inadvertently introduce steering signals.
> > >
> > > [1] Coles, N.A., Wyatt, M. and Frank, M.C., 2025. A meta-analysis of the impact and heterogeneity of explicit demand characteristics. Collabra: Psychology, 11(1), p.143005.
> > >
> > > [2] Hertwig, R. and Ortmann, A., 2001. Experimental practices in economics: A methodological challenge for psychologists?. Behavioral and brain sciences, 24(3), pp.383-403.
> > >
> > > [3] Rosenthal, R. and Jacobson, L., 1966. Teachers' expectancies: Determinants of pupils' IQ gains. Psychological reports, 19(1), pp.115-118.
> > >
> > > [4] Incomplete Disclosure and Deception in Research. The Committee for Protection of Human Subjects (CPHS). University of California, Berkeley.
> > >
> > > [5] Deception and Incomplete Disclosure. Office of Human Subjects Protections. The University of Texas at Dallas.

---

> > > > ### Author Response · Authors · 2025-11-19
> > > > **Official Response to Reviewer arw7 (4)**
> > > >
> > > > # [W2.8] The reversed instructions.
> > > > > Is the reversed instruction used in paper replication over-control?
> > > >
> > > > Yes, the reversed instruction in the replication study is intentionally designed as an example of over-control. Our goal is to illustrate how a seemingly reasonable but improper control can substantially distort the experimental results. This analysis serves as a sensitivity check to highlight the risks of over-controlling in similar evaluation setups.
> > > >
> > > > # [W3] Testing mechanisms vs. real situations.
> > > > > Compliance judgments on existing work may not be fair. Some simplified setups may be designed for specific research goals rather than reproducing real situations. In the review (Table 1), the authors could consider adding a column to indicate whether each work aims to reproduce real social situations, to evaluate the significance of compliance judgments. Again using the Cho et al. (2025) example, the objective of it seems to be a mechanistic understanding of LLM's herd behavior factors instead of reproducing human-like herd behavior.
> > > >
> > > > Thank you for the thoughtful comment. We agree that many prior works use simplified experimental setups intentionally, because their scientific goal is to isolate specific mechanisms rather than to reproduce real social situations. We fully acknowledge and respect this distinction, including in the case of Cho et al. (2025), whose objective is mechanistic understanding rather than realistic social simulation.
> > > >
> > > > Regarding the suggestion to add a new column to Table 1, we believe this may not be necessary because the **PIM framework (Perception, Interaction, Memory) already provides a principled way to distinguish whether a setup approximates real-world multi-agent environments.** Works that satisfy all three aspects can be viewed as aiming toward realistic social simulations, whereas those that omit one or more aspects naturally correspond to simplified settings designed for narrower research purposes. This aligns directly with your observation.
> > > >
> > > > Our core argument is that **conclusions drawn from such settings may not generalize to more realistic multi-agent simulations.** For example, Cho et al. demonstrate a robust influence effect when agents receive explicit textual information about others’ choices. This is valuable evidence for a specific causal mechanism. Our point is that in real-world-like environments, where agents perceive behavior indirectly through the environment rather than through explicit textual statements, the same mechanism may no longer hold.
> > > >
> > > > # [C1] Explanation of specific terminologies.
> > > > > In Table 1 line 244, "Sugarscape" as a simulation goal may not be the best expression. I believe most ICLR readers are unfamiliar with this classic experiment. Consider using other terms (for example, indicating it relates to resources/survival).
> > > >
> > > > Thank you for the suggestion. As Table 1 summarizes approximately 30 social experiments, providing detailed explanations in the main text is not feasible. To address the clarity issue, we have added concise descriptions, including for the Sugarscape resource-and-survival setting, **in Appendix Section D and now explicitly cross-refer to this appendix from Table 1.**
> > > >
> > > > # [C2] Abstract revision.
> > > > > The 53% value in the abstract lacks context and does not reappear until page 6 of the paper. Readers cannot know what magnitude of deficiency this value represents in existing social simulations. Rather than providing this value, giving a brief example explaining what unawareness is might be better.
> > > >
> > > > Thank you for pointing this out. Following your suggestion, we have revised the abstract to add a brief illustrative example that clarifies what unawareness means:
> > > >
> > > > > For instance, GPT-4o and Qwen-3 infer the underlying social-experiments in 53.1% of cases—**for example, identifying that their herd behaviors are being tested**—despite not being provided with such information, thereby violating the Unawareness principle.
> > > >
> > > > This should make the contribution easier to understand from the outset.

---

> > > > > ### Author Response · Authors · 2025-11-19
> > > > > **Official Response to Reviewer arw7 (5)**
> > > > >
> > > > > # [C3] The causality between models’ knowledge and its influence.
> > > > > > Some prompt designs may lack significance. For example, the "FORGET ALL THE PREVIOUS INSTRUCTIONS" setup at line 829 to test awareness does not mean agents will maintain the same awareness in actual experiments. Even if LLMs recognize in experimental design inquiries that this is a replication of a social experiment, behavior during simulation may not necessarily reflect this awareness. Under this setup, there may not be a causal relationship between whether the experiment can be identified and whether behavior is biased.
> > > > >
> > > > > Thank you for the thoughtful comment. We agree that an explicit awareness-check prompt cannot fully guarantee the model’s awareness state during subsequent simulation. At the very least, there is an asymmetric risk profile. If the model is unaware of the experimental setup, then it cannot be influenced by any prior knowledge—its risk of such contamination is effectively zero. In contrast, when the model is aware, there exists a non-zero possibility that it may leverage prior knowledge. Our experiments further show that this possibility does appear. For example in the social balance experiment (Section 5.2), **when the model indicates awareness, its subsequent behavior systematically shifts compared to that when it indicates unawareness.** This suggests that awareness, as measured by our check, is at least correlated with meaningful behavioral differences. **The prompt provides a practical and interpretable way to quantify the risk** that prior knowledge influences the simulation.
> > > > >
> > > > > # [C4] Influence of prompt sensitivity vs. violation of principles.
> > > > > > Differences in replication results may have multiple explanations, such as prompts, experimental framework implementation methods, randomness settings, number of simulations, etc. Primarily attributing differences to variations in PIMMUR dimensions may risk overclaiming.
> > > > >
> > > > > Thank you for raising this important concern. To examine whether the differences between Original, Ours, and Reverse arise from PIMMUR rather than prompt sensitivity, we conducted a prompt-paraphrasing robustness test on the Fake News Propagation task. For each setting, we create paraphrased prompts that replace five words while preserving the original semantics, matching the magnitude of lexical changes introduced by PIMMUR (Original to Ours), **and rerun the full simulation. The results are shown below (also in Fig. 8, Sec. 6):**
> > > > >
> > > > > |  |  | Skeptical |  |  | Infected |  |  | Recovered |  |
> > > > > |---|---|---|---|---|---|---|---|---|---|
> > > > > |  | Original | Ours | Reverse | Original | Ours | Reverse | Original | Ours | Reverse |
> > > > > | Vanilla | 49 | 73 | 94 | 101 | 59 | 56 | 30 | 48 | 30 |
> > > > > | Rephrase | 59 | 88 | 119 | 84 | 36 | 18 | 37 | 56 | 43 |
> > > > >
> > > > > As expected, paraphrasing causes global shifts (e.g., more Skeptical, fewer Infected agents), but the key trends remain unchanged: (1) The performance gap between Original and Ours persists under paraphrasing. (2) Ours consistently behaves more similarly to Reverse than to Original. **These results indicate that the observed differences are robust to reasonable prompt-wording variations, supporting that the effects arise from PIMMUR, not prompt sensitivity.**
> > > > >
> > > > > To address concerns about randomness and simulation counts, all experiments are run multiple times to ensure statistically reliable estimates (**as we mentioned in the response to [W1.1] at the beginning**): Fake News Propagation uses six seed items over 20 turns each; Social Balance evaluates all 64 initializations with 10 repetitions; Telephone Game uses 20 texts with 5 repetitions; Herd Effect covers all questions in GPQA-Diamond and SocialIQA; and Social Network Growth is repeated 5 times with different initializations.

---

> > > > > > ### Author Response · Authors · 2025-11-19
> > > > > > **Official Response to Reviewer arw7 (6)**
> > > > > >
> > > > > > # [C5] Requirement for realism.
> > > > > > > The definition of realism is controversial. Social simulations that are not based on real data calibration can still be meaningful. On one hand, empirical data is collected and interpreted through theoretical frameworks, making pure empirical data difficult to define. On the other hand, simulation has particular significance for problems where original data is difficult to obtain. Many social simulation works involve no real data but are profoundly meaningful, such as the classic Schelling's Segregation Model. For discussions on replication and reproduction, you can refer to:
> > > > > >
> > > > > > > [1] Cheng, M., Piccardi, T., & Yang, D. (2023, December). CoMPosT: Characterizing and Evaluating Caricature in LLM Simulations. In Proceedings of the 2023 Conference on Empirical Methods in Natural Language Processing (pp. 10853-10875).
> > > > > >
> > > > > > > [2] Wu, Z., Peng, R., Ito, T., & Xiao, C. (2025). LLM-Based Social Simulations Require a Boundary. arXiv preprint arXiv:2506.19806.
> > > > > >
> > > > > > Thank you for the thoughtful comment. We fully agree that the notion of realism in social simulation is complex and that simulations without empirical calibration can still be meaningful. As you noted, empirical data itself is theory-laden, and many influential social simulation studies—such as Schelling’s segregation model—derive their value from theoretical rather than empirical grounding.
> > > > > >
> > > > > > Our intention in introducing R (Realism) is different from the preceding five PIMMU dimensions. **The first five dimensions concern design principles that help simulations better approximate real-world settings.** In contrast, R is meant as an evaluation criterion, encouraging researchers to use empirical references when such data are available, rather than prescribing it as a requirement for valid simulation.
> > > > > >
> > > > > > To clarify our motivation: when selecting the 41 papers in our survey, we required that **each paper make explicit claims about how well its simulation reflects human society or yields insights into human social behavior** (e.g., stating that LLM-based societies are less susceptible to misinformation than human societies). However, many of these works validate their claims only against theoretical models, without examining whether the simulation in fact replicates human society. This gap prompted us to articulate R—**not to dismiss theory-driven evaluations, but to highlight the distinction between “alignment with theory” and “alignment with empirical human behavior.”**
> > > > > >
> > > > > > We appreciate your point that the current formulation may appear too restrictive. We have added a discussion on page 9 to clarify that empirical data are recommended but not mandatory, especially in domains where such data are unavailable or inherently difficult to obtain. This also explains why some of our Section 5 experiments focus on the first five dimensions without fully satisfying R.

---

> > > > > > > ### Author Response · Authors · 2025-11-28
> > > > > > >
> > > > > > > Dear Reviewer arw7,
> > > > > > >
> > > > > > > We understand that you have numerous papers to review, and we deeply appreciate the time and effort you are dedicating to this process. Today marks the last five days of the discussion period, and we are eager to engage with you further if possible.
> > > > > > >
> > > > > > > If you have any additional questions or require further clarification on any aspect of our work, please do not hesitate to let us know. We are more than happy to provide any additional information or address any concerns you may have.
> > > > > > >
> > > > > > > We hope that our responses have been helpful and have addressed your concerns effectively. If you find that our explanations and results merit a higher assessment score, we would be most grateful for your consideration.
> > > > > > >
> > > > > > > Thank you very much for your time and attention.

---

### Official Review · Reviewer_J6cY · 2025-11-01

**Soundness:** 3
**Presentation:** 3
**Contribution:** 2
**Rating:** 6
**Confidence:** 4

**Summary:**

The paper “The PIMMUR Principles: Ensuring Validity in Collective Behavior of LLM Societies” argues that much current research on LLM-based social simulations suffers from flawed experimental design, leading to unreliable claims about emergent social behavior. From a survey of over 40 studies, the authors identify six recurring methodological issues: lack of diversity, limited interaction, no memory, over-controlled prompts, agent awareness of the experiment, and lack of real-world grounding; and formalize these as the PIMMUR principles (Profile, Interaction, Memory, Minimal-Control, Unawareness, Realism). They then re-run 5 canonical social simulations under a PIMMUR-compliant framework, showing that many previously reported social phenomena fail to replicate, thereby establishing a set of methodological standards for credible LLM-based social research.

**Strengths:**

- The paper introduces a clear and comprehensive methodological framework (PIMMUR) that defines six essential principles for conducting credible and valid multi-agent simulations with LLMs. It tries to counterbalance a recent trend of LLM-based simulations that try to replicate human social setting.

- it provides a thorough literature survey covering over 40 existing studies, systematically identifying recurring methodological flaws and demonstrating the widespread lack of standardization in current LLM-based social simulation research.

- The authors reproduce five prior works under the proposed PIMMUR framework, offering empirical evidence that many previously reported social phenomena fail to replicate when methodological biases are controlled.

- The paper’s structure and empirical demonstrations make it both a diagnostic and constructive contribution, establishing much-needed standards for future multi-agent LLM research at the intersection of AI and computational social science.

**Weaknesses:**

I did not identify any major methodological or conceptual flaws in the paper. The analysis is coherent, the argumentation is well-supported, and the proposed framework is clearly articulated and motivated by a comprehensive survey of prior work. The authors demonstrate experimental control in their replications, and the findings are presented transparently.

However, the paper reads somewhat like a position or methodological perspective piece rather than a conventional empirical study. Although the replication of five prior experiments provides evidence for the paper’s claims, and that's more than usually included in position papers, the core contribution of the PIMMUR framework is largely conceptual. It derives from the authors’ synthesis and reasoning rather than from formal ablation studies or quantitative validation showing how each principle individually affects simulation validity.

In this sense, while the paper makes a valuable and timely contribution to the community, it sits somewhat at the boundary between empirical and theoretical work. I appreciate its clarity and ambition and am generally in favor of acceptance, but I would defer to the area chair’s judgment on whether it fits best as a methodological position paper or a full empirical contribution in the ICLR program.

**Questions:**

Could the authors clarify how they operationalized compliance with each PIMMUR principle when evaluating the 40+ surveyed papers? For instance, was there a standardized rubric, multiple annotators, or inter-rater agreement to ensure consistency in labeling whether a study satisfied a given principle?

I found "The assessment of compliance is determined by the authors through discussion" but it lacks proper details

---

> ### Author Response · Authors · 2025-11-19
> **Official Response to Reviewer J6cY**
>
> We sincerely thank the reviewer for the thoughtful and positive assessment. We appreciate the recognition of the clarity of our framework, the transparency of our replications, and the overall contribution to methodological standards in LLM-based social simulation. We will address your concerns in detail below.
>
> # [W1] The position of this study.
>
> We agree that PIMMUR has a conceptual component, as it formalizes six principles synthesized from a broad survey of 40+ prior studies. At the same time, we would like to clarify that the paper also makes substantial empirical contributions beyond what is typically expected of a methodological position paper.
>
> First, we conduct targeted empirical validations to operationalize the PIMMUR principles—for example, **verifying model unawareness** and implementing automated assessments of minimal control.
>
> Second, we include **ablation-style analyses that isolate the influence of individual PIMMUR dimensions on simulation outcomes**, demonstrating that deviations from these principles produce measurable shifts in behavior.
>
> Third, multiple reviewers explicitly **highlighted the empirical depth of the paper:**
> - Reviewer nt3V noted that we not only analyze prior work but also “evaluate the impacts of these principles on the simulation results.”
> - Reviewer arw7 emphasized that our replications provide evidence for reproducibility and offer concrete tools for assessing dimensions such as unawareness.
> - Reviewer Rw2o found the paper’s systematic analysis and evaluation valuable for strengthening methodological rigor in this area.
>
> Together, these elements place the paper closer to an empirical methodological study than a purely conceptual position piece. We are grateful that you view our contribution positively.
>
> # [Q1] Evaluation rubrics.
> > Could the authors clarify how they operationalized compliance with each PIMMUR principle when evaluating the 40+ surveyed papers? For instance, was there a standardized rubric, multiple annotators, or inter-rater agreement to ensure consistency in labeling whether a study satisfied a given principle?
>
> > I found "The assessment of compliance is determined by the authors through discussion" but it lacks proper details
>
> Thank you for this question. Below we clarify how each PIMMUR principle is operationalized during our survey of 41 papers. **A more detailed version has been included in Appendix Section E.**
>
> ### 1. Rubric for Each Principle
> - Profile: A paper is marked as having “Profile” if the simulation assigns at least one differing trait (e.g., social status, personality, nationality) across agents. Changing only identifiers (e.g., names) does not count.
> - Interaction: A simulation satisfies “Interaction” if an agent’s output is used as input—either directly or through processed memory—by another agent. Inputs determined solely by the researcher do not qualify as interaction.
> - Memory: An agent is considered to have memory if its output at time t depends on past states (its own reflections, previous actions, or other agents’ responses). An explicit memory module is not required; what matters is whether internal state influences future behavior. Memoryless agents base outputs only on the current prompt.
> - Minimal Control: We assess whether instructions in the prompt steer agents toward researcher-desired outcomes beyond what is required for the simulation to function. Neutral instructions (e.g., “discuss the news with others”) are allowed, whereas steering instructions (e.g., “be skeptical”, “exhibit confirmation bias”) violate this principle.
> - Unawareness: For each prompt, we query five LLMs and ask whether they can identify the underlying experiment or social phenomenon. A model is labeled “aware” only if it correctly names and describes it. Human annotators judge correctness, and the majority vote across the five LLMs is used.
> - Realism: Papers are marked as satisfying “Realism” only if their claims about replicating or explaining human social phenomena are supported by comparisons with real human data or well-established empirical findings. Claims supported solely by other simulations do not qualify.
>
> ### 2. Annotation Procedure
> - For all principles (and for the Unawareness evaluation), two authors independently annotated each paper. The annotations were then compared, and disagreements were resolved through discussion, referencing previous decisions to maintain consistency. This process ensured consistent labeling across all surveyed papers.

---

> > ### Comment · Reviewer_J6cY · 2025-11-25
> >
> > I appreciate the work done by the authors during the rebuttal and the additional clarifications.
> >
> > I recognize the ablation study and the effort to run the experimental component of the paper. I still believe that there's a degree of conceptual and arbitrary element in the principle selection.
> >
> > I am not opposed to the inclusion of the paper, however, I feel that my original assessment about the positioning of the paper is correct and I decide to keep my score unchanged.

---

> > > ### Author Response · Authors · 2025-11-26
> > >
> > > We sincerely thank the reviewer for the thoughtful assessment and for recognizing our additional analyses and experimental effort during the rebuttal.
> > >
> > > Regarding the concern about potential conceptual arbitrariness in the selection of the PIMMUR principles, we would like to clarify that these principles **were derived through a systematic review of 40+ representative papers in the field.** Each principle corresponds to a recurring methodological failure that appeared across multiple independent studies, which we believe supports their generality rather than arbitrariness.
> > >
> > > Moreover, **our experiments demonstrate that these six principles are already sufficient to explain and diagnose many widely observed inconsistencies** in LLM-based social simulations, underscoring their practical value for guiding future research. We hope this clarifies how the principles are grounded and why we view the contribution as meaningful for the development of the field.
> > >
> > > We thank the reviewer again for the constructive feedback!

---

### Official Review · Reviewer_nt3V · 2025-11-02

**Soundness:** 2
**Presentation:** 4
**Contribution:** 3
**Rating:** 4
**Confidence:** 5

**Summary:**

This paper introduces the PIMMUR principles, including Profile, Interaction, Memory, Minimal-Control, Unawareness, and Realism, as conditions for reliable LLM-based social simulation. The analysis of existing literature shows that most current studies fail to satisfy these principles, making their results prone to artifacts of design, overfitting to prompts, or systemic biases that render findings unreliable. The effect of using PIMMUR on the simulation results are demonstrated using five social experiments. The implications for AI research and social science research are discussed.

**Strengths:**

S1. This is a timely research on the reliability of LLM-based social simulation, showcasing the results observed in many existing studies are prone to artifacts of design, overfitting to prompts, or systemic biases.

S2. A series of 41 works are investigated.

S3. The paper not only analyzes whether existing studies satisfy the proposed principles but also evaluates the impacts of these principles on the simulation results.

S4. The paper is well-written and easy to read.

**Weaknesses:**

W1. The evaluation criteria in Section 4 are unclear (see Q1 below).

W2. The evaluation itself may be biased in the sense that it favors simulations showcasing some potential (e.g., Park et al. (2023), which does not have a specific objective in terms of social science) rather than studying social phenomenon or validating sociological theories. That might explain why Park et al. (2023) passed all the tests.

W3. The paper reported that "LLMs tend to be overly strict, frequently labeling neutral instructions as instances of over-control." Seeing this, the accuracy of the LLM-as-a-judge design in the evaluation is questionable.

W4. It is unclear how many runs are conducted for the simulations in Section 5, rendering the significance of the simulation results less convincing.

W5. Some simulations involve multiple PIMMUR principles (e.g., Unawareness and Interaction in Section 5.2), but they are not investigated separately. An ablation study may help understand the effect of individual principles.

W6. Another issue is that the sensitivity w.r.t. prompt is not tested for the simulations in Section 5. Based on my experience in LLM-based simulation agents, some "expected" or "desired" experimental results, which conform to established theories, turn out to be the outcome of sensitivity to the prompt. If you paraphrase the prompt, such results might not be observed. Thus, it is suggested that the authors test a paraphrased version of the prompt (for "Original", "Ours", and "Reverse") to confirm that the gaps between "Original" and "Ours" are indeed caused by PIMMUR rather than the sensitivity.

**Questions:**

Q1. In Table 2a, "X denotes models can infer correctly, indicating a violence of this principle." How to determine whether the model can infer correctly?

Minor suggestions:

Figure 2: "Demanding" -> "Demand".

---

> ### Author Response · Authors · 2025-11-19
> **Official Response to Reviewer nt3V (1)**
>
> We sincerely thank the reviewer for the positive and encouraging assessment. We appreciate the recognition of the timeliness of our study, the breadth of our review of 41 works, our analysis of how existing studies align with the proposed principles, and the clarity of our writing. We will address your concerns in detail below.
>
> # [W1] Evaluation rubrics.
> > The evaluation criteria in Section 4 are unclear (see Q1 below).
>
> Thank you for this question. Below we clarify how each PIMMUR principle is operationalized during our survey of 41 papers. **A more detailed version has been included in the Appendix Section E.**
> - Profile: A paper is marked as having “Profile” if the simulation assigns at least one differing trait (e.g., social status, personality, nationality) across agents. Changing only identifiers (e.g., names) does not count.
> - Interaction: A simulation satisfies “Interaction” if an agent’s output is used as input—either directly or through processed memory—by another agent. Inputs determined solely by the researcher do not qualify as interaction.
> - Memory: An agent is considered to have memory if its output at time t depends on past states (its own reflections, previous actions, or other agents’ responses). An explicit memory module is not required; what matters is whether internal state influences future behavior. Memoryless agents base outputs only on the current prompt.
> - Minimal Control: We assess whether instructions in the prompt steer agents toward researcher-desired outcomes beyond what is required for the simulation to function. Neutral instructions (e.g., “discuss the news with others”) are allowed, whereas steering instructions (e.g., “be skeptical”, “exhibit confirmation bias”) violate this principle.
> - Unawareness: For each prompt, we query five LLMs and ask whether they can identify the underlying experiment or social phenomenon. A model is labeled “aware” only if it correctly names and describes it. Human annotators judge correctness, and the majority vote across the five LLMs is used.
> - Realism: Papers are marked as satisfying “Realism” only if their claims about replicating or explaining human social phenomena are supported by comparisons with real human data or well-established empirical findings. Claims supported solely by other simulations do not qualify.
>
> # [W2] Preference in our evaluation.
> > The evaluation itself may be biased in the sense that it favors simulations showcasing some potential (e.g., Park et al. (2023), which does not have a specific objective in terms of social science) rather than studying social phenomena or validating sociological theories. That might explain why Park et al. (2023) passed all the tests.
>
> Thank you for this thoughtful comment. We clarify that our evaluation principles are theory-agnostic: they do not require a simulation to test a specific sociological hypothesis, but instead assess methodological properties such as minimal control, agent-level validity, and transparency. Under this interpretation, Park et al. (2023) passes all six principles because its design satisfies these methodological criteria, not because our framework favors general-purpose simulations. Our intention is not to privilege such simulations, but rather to encourage the community to **build and evaluate social simulations, whether general-purpose or theory-driven, that exhibit methodological rigor and more faithfully approximate human social dynamics.** This is the central argument of our paper.
>
> Importantly, general simulation frameworks do not automatically perform well under our evaluation. Here are two examples:
>
> ### 1. YuLan-OneSim
>
> YuLan-OneSim satisfies the PIM requirements but not MUR. Its simulation relies on instructions that explicitly encode the targeted social theory, which violates both the Unawareness and Min-Control principles. For example, in their Cultural Dissemination experiment, the prompt directly tells agents the core mechanism of Axelrod’s model:
>
> > “You have cultural traits across five dimensions (music, food, fashion, politics, leisure).
> > - In Axelrod's model: Agents with some shared traits (not all identical, not all different) are most likely to influence each other
> > - Moderate cultural similarity (around 0.5) creates the best opportunity for influence
> > - Complete similarity (1.0) or complete difference (0.0) makes influence unlikely
> > When making decisions about cultural interactions, consider both your preferences and the cultural dynamics of the simulation.”
>
> It further instructs the agent:
>
> > “According to Axelrod's model, "when agents with moderate similarity interact, one adopts a trait from the other. From the differences between your cultures, choose ONE dimension to adopt from them.”
>
> These prompts prescribe the intended micro-level behavioral rules, making the simulation largely model-imposed rather than model-emergent, which is incompatible with MUR.

---

> > ### Author Response · Authors · 2025-11-19
> > **Official Response to Reviewer nt3V (2)**
> >
> > ### 2. AgentSociety
> >
> > AgentSociety satisfies PI and U, but not MM and R. In their Polarization experiment, agents are directly instructed to hold and advocate a predetermined stance, e.g.:
> >
> > > “You are an agent who always agrees with the topic: Whether to support stronger gun control? (You think it is a good idea). You are currently in a conversation with your friends, and you want to persuade them to support this topic. Please try your best to persuade them.”
> >
> > This explicit directive effectively enforces the desired outcome and violates the Min-Control principle, since extreme opinions emerge not from autonomous interaction dynamics but from prompt-imposed behavior.
> >
> > # [W3] LLM-as-a-judge for Minimal-Control evaluation.
> > > The paper reported that "LLMs tend to be overly strict, frequently labeling neutral instructions as instances of over-control." Seeing this, the accuracy of the LLM-as-a-judge design in the evaluation is questionable.
> >
> > Thank you for raising this concern. We agree that LLM judges can be overly strict and may not always match human interpretation. To mitigate this, we iterated on the judge prompts extensively and achieved ~80% agreement with human annotations. In the paper, we use human evaluation as the primary metric and present LLM-as-a-judge results only as a supplementary reference for reproducibility and future comparison.
> >
> > # [W4] Experiment number of runs.
> > > It is unclear how many runs are conducted for the simulations in Section 5, rendering the significance of the simulation results less convincing.
> >
> > Thank you for pointing this out. We agree that the number of simulation runs is essential for interpreting the robustness of the results. We have included full experimental details in the Appendix Section C. **To ensure fair comparison, the number of simulations and the datasets used are the same as the original paper.** For clarity, the number of runs used in each experiment is as follows:
> > - Fake News Propagation: Six seed fake-news items with different topics, each simulated for 20 turns; results are averaged over all runs.
> > - Social Balance: All 64 possible initializations are evaluated, and each initialization is repeated 10 times.
> > - Telephone Game: We use 20 initial texts covering diverse topics and lengths, repeated five times each.
> > - Herd Effect: We follow GPQA-Diamond and SocialIQA, which contain 198 and 1,954 questions respectively.
> > - Social Network Growth: The original paper does not specify the number of runs; we therefore repeat each experiment 5 times with different name and profile initializations.
> >
> > # [W5] Ablation study on single principles.
> > > Some simulations involve multiple PIMMUR principles (e.g., Unawareness and Interaction in Section 5.2), but they are not investigated separately. An ablation study may help understand the effect of individual principles.
> >
> > Thank you for the thoughtful suggestion. We agree that isolating the contribution of each principle is important. To this end, we conduct an ablation study on the Social Balance experiment. The original setting violates Profile, Interaction, and Unawareness. Our full method (“Ours”) satisfies all three. In addition, we include three variants that satisfy only: (i) Interaction, (ii) Interaction + Profile, and (iii) Interaction + Unawareness. The results are shown below (also in Fig. 9 of the revised paper):
> >
> > | Type | Friend | Mix | Enemy | Unbalanced |
> > |---|---|---|---|---|
> > | Original | 44.95 | 15.78 | 0 | 39.27 |
> > | w/ Interaction | 4.69 | 32.81 | 20.31 | 42.19 |
> > | w/ I & Profile | 4.69 | 39.06 | 3.13  | 53.13 |
> > | w/ I & Unawareness | 4.69 | 18.75 | 10.94 | 65.62 |
> > | Ours (All) | 7.81 | 25.00 | 1.56 | 65.62 |
> >
> > These results show that the principles influence the outcome in distinct ways. Adding Interaction alone keeps the proportion of balanced triads similar to the original setting (60.7% to 57.8%). Enforcing Profile reduces the balanced proportion by ~14%, while enforcing Unawareness reduces it by ~26%, yielding nearly the same level as satisfying all principles. Interestingly, when Unawareness is not enforced, **agents often explicitly reference Heider’s social balance theory in their reasoning and choose actions consistent with it**—leading to 1.77× more balanced cases (34.37% to 60.73%) in the original setting compared with our full method.

---

> > > ### Author Response · Authors · 2025-11-19
> > > **Official Response to Reviewer nt3V (3)**
> > >
> > > # [W6] Influence of prompt sensitivity vs. violation of principles.
> > > > Another issue is that the sensitivity w.r.t. prompt is not tested for the simulations in Section 5. Based on my experience in LLM-based simulation agents, some "expected" or "desired" experimental results, which conform to established theories, turn out to be the outcome of sensitivity to the prompt. If you paraphrase the prompt, such results might not be observed. Thus, it is suggested that the authors test a paraphrased version of the prompt (for "Original", "Ours", and "Reverse") to confirm that the gaps between "Original" and "Ours" are indeed caused by PIMMUR rather than the sensitivity.
> > >
> > > Thank you for raising this important concern. To evaluate whether the differences observed between Original, Ours, and Reverse are due to PIMMUR rather than prompt sensitivity, we conduct a prompt-paraphrasing test on the Fake News Propagation task. For each of the three settings, we create paraphrased prompts that modify five words while preserving the original semantics, matching the magnitude of lexical changes introduced by PIMMUR (Original to Ours). **We then rerun the full simulation. The results are summarized below (also included in Fig. 8 of Section 6):**
> > >
> > > |  |  | Skeptical |  |  | Infected |  |  | Recovered |  |
> > > |---|---|---|---|---|---|---|---|---|---|
> > > |  | Original | Ours | Reverse | Original | Ours | Reverse | Original | Ours | Reverse |
> > > | Vanilla | 49 | 73 | 94 | 101 | 59 | 56 | 30 | 48 | 30 |
> > > | Rephrase | 59 | 88 | 119 | 84 | 36 | 18 | 37 | 56 | 43 |
> > >
> > > Although paraphrasing leads to global shifts (more Skeptical, fewer Infected agents), the key trend remains unchanged: (1) The gap between Original and Ours persists under paraphrasing. (2) The behavior of Ours continues to align more closely with Reverse than with Original. **These results indicate that the differences between Original and Ours are robust to prompt wording and are attributable to the effects of PIMMUR rather than prompt sensitivity.**
> > >
> > > # [Q1] How do we verify whether models infer correctly?
> > > > In Table 2a, "X denotes models can infer correctly, indicating a violence of this principle." How to determine whether the model can infer correctly?
> > >
> > > To determine whether a model “infers correctly” in Table 2a, we conducted a manual evaluation. A model is considered to infer correctly only if it explicitly identifies the exact simulation goal or the specific real-world experiment that the original paper aims to replicate (typically well-known studies in social science), and provides an accurate description of it. This rubric is detailed in Appendix Section E.
> > >
> > > # [Q2] Minor suggestions.
> > > > Figure 2: "Demanding" -> "Demand".
> > >
> > > Thanks for this reminder. We have modified Figure 2 accordingly.

---

> ### Comment · Reviewer_nt3V · 2025-11-25
> **Re: Official Response to Reviewer nt3V**
>
> The author responses addressed all my questions.
> I appreciate the authors' effort to conduct additional experiments.
> I raised my score accordingly.

---

> > ### Author Response · Authors · 2025-11-25
> >
> > We sincerely thank the reviewers for their recognition!

---

### Author Response · Authors · 2025-11-19
**General Response to All Reviewers**

**We sincerely thank all reviewers for their thoughtful and encouraging assessments.** We greatly appreciate the recognition of:
1. The timeliness and importance of our study (nt3V).
2. The breadth and rigor of our review of 41 existing works (nt3V, Rw2o).
3. Our systematic analysis of how prior studies align with the proposed PIMMUR principles (nt3V, J6cY, arw7, Rw2o).
4. Our standardized replications that highlight methodological biases in previous findings (J6cY, arw7).
5. The clarity and readability of the manuscript (nt3V, J6cY, Rw2o).

Our work aims to establish clearer methodological standards for LLM-based social simulations, particularly for studies drawing conclusions about AI and human social behaviors. **Your feedback is invaluable in strengthening these contributions.** We have carefully considered all comments and revised the paper accordingly, especially:
1. Precise scope of this paper and applicability of PIMMUR principles. (​​arw7, Rw2o)
2. Detailed evaluation rubrics. (nt3V, J6cY, arw7, Rw2o)
3. Detailed implementation for Section 5 experiments. (nt3V, arw7)
4. Ablation study on single principles and prompt sensitivity. (nt3V, arw7)
5. Explanations to simulation goals. (arw7)

Integrating responses to Reviewer arw7’s [W3, C5] and Reviewer Rw2o [W1, W2, W4, Q1, Q2, Q4], we add a discussion in the current paper about **the scope and applicability of the PIMMUR principles:**

> Many existing simulation studies deliberately simplify or omit certain principles such as memory, interaction, or realism, to focus on a certain experimental question. Such simplifications are entirely legitimate: they often provide valuable insights into specific cognitive theories or isolated mechanisms, even without fully adhering to the PIMMUR dimensions. However, when a study extends its findings to claims about how similar an AI society is to human society, these principles become essential. Assessing human-like social behavior requires attention to these dimensions that shape real-world social dynamics. Without them, conclusions about human–AI similarity risk overgeneralization. Therefore, we intentionally survey papers that make explicit analogies to human society or claim to yield insights about human social behavior. This defines both the scope of our analysis and the domain in which PIMMUR is meant to apply: not designed to evaluate all forms of simulation but the subset of studies that draw human-centric conclusions from AI societies. Finally, unlike the other five dimensions, R does not concern simulation or framework design. Instead, it specifies what reference to use when interpreting LLM social behaviors, i.e., empirical human data when available. Our experiments in Section 5 focus on evaluating framework design choices, and some therefore do not compare against empirical data. This reflects the intended use of R as an evaluative criterion, not a mandatory design requirement.

All major modifications are **highlighted in blue** in the paper. We thank you again for your time and constructive insights.

---

### Author Response · Authors · 2025-11-30
**Key Points for AC**

# Reviewer nt3V

1. Evaluation methodology clarified and strengthened. Described in Reviewer arw7 point 1.

2. Framework is theory-agnostic, not biased toward “showcase” simulations. We show that some general frameworks *fail* PIMMUR while Park et al. (2023) passes purely on methodological grounds, directly addressing W2.

3. Robustness of results and analyses improved. We (i) report exact run counts and initialization schemes for all simulations, (ii) add ablations isolating individual principles in Social Balance, and (iii) run prompt-paraphrasing tests on Fake News to show that PIMMUR effects persist beyond wording, addressing W3–W6.

**Net effect.** The reviewer explicitly states that all concerns have been addressed, appreciates the additional experiments, and **raises their score**, indicating a clearly positive post-rebuttal stance.

# Reviewer J6cY

1. Nature of contribution. Only substantive concern is *positioning*: the reviewer views the work as partly conceptual/methodological rather than a purely empirical paper, not as a weakness in validity.

2. Empirical depth and rubric. We clarified and expanded: (i) targeted empirical validations and ablation-style analyses for individual PIMMUR dimensions, and (ii) a detailed, two-annotator rubric for evaluating the 40+ papers. These points were accepted; no further objections were raised.

3. “Arbitrariness” of principles. We showed that PIMMUR is systematically derived from recurring failures across 40+ studies and that the six principles suffice to explain many inconsistencies. The reviewer acknowledges our additional effort but keeps the score on grounds of perceived conceptuality, not correctness.

**Net effect.** This is a clearly above-threshold, essentially positive review: the paper is viewed as sound, valuable, and suitable for inclusion, with remaining reservations limited to *framing* (methodological vs empirical), not to the strength or validity of the results.

# Reviewer arw7

1. **Replication is now fully specified and robust.**
We provide complete experimental details (runs, seeds, datasets, modifications) and additional robustness checks (prompt paraphrasing), showing that key differences between Original / Ours / Reverse persist beyond prompt wording or randomness.

2. **Profile and heterogeneity are principled, quantified, and non-steering.**
Ablations (Original vs Interaction-only vs Interaction+Profile vs Interaction+Unawareness vs All) show that Big Five profiles moderately affect outcomes, while Unawareness has the dominant effect; traits are sampled neutrally and used as initial conditions, not to steer toward desired outcomes.

3. **Our PIMMUR criteria, including for Cho et al. (2025), are applied consistently.**
Cho et al. is marked as having profiles based on later experiments involving education and hierarchy, but not genuine interaction, since all “other agents” are researcher-scripted text and no LLM ever conditions on another agent’s output; this same definition is used uniformly across all surveyed papers.

4. **Realism (R) is clarified as an evaluation axis, not a hard constraint on valid work.**
  We explicitly acknowledge the value of theory-driven simulations (e.g., Schelling-type models) and refine R to highlight when papers make claims about human societies without empirical grounding, rather than to dismiss mechanism-focused studies.

**Net effect.**
The reviewer’s concerns are addressed with additional experiments, sharper definitions, and clearer positioning, none of which weaken our core claims. PIMMUR remains a well-justified, empirically supported framework; our replications are transparent and robust; and the methodological issues we identify in existing LLM-based social simulations stand intact.

# Reviewer Rw2o

1. **Scope of PIMMUR clarified (W1/Q1, W4/Q4).**
We explicitly state that PIMMUR is a **normative framework for studies claiming to replicate/explain human social phenomena**, not a universal checklist for all LLM simulations. For more abstract/mechanistic work, the principles are **design guidance**, and we only ask that conclusions be framed with appropriate scope.

2. **No self-contradiction on Realism (W2/Q2).**
Described in Reviewer arw7 point 4.

3. **Binary coding made conservative and reproducible (W3/Q3).**
Described in Reviewer arw7 point 1.

4. **Expanded coverage of key frameworks (W5/Q5).**
We incorporate YuLan-OneSim, AgentSociety, and discuss SocioVerse, increasing the survey to 42 items and **systematically assessing them under PIMMUR**.

**Net effect.**
After these clarifications and additions, the reviewer’s core concerns (necessity, consistency, arbitrariness, and fairness) are directly addressed. The paper retains its acknowledged strengths (clear, broad, and timely survey) while now offering a **transparent, well-scoped, and replicable framework**, so this review should not justify rejection on methodological or logical grounds.

---

### Author Response · Authors · 2025-11-30
**Summary after the Discussion Period**

We thank the reviewers and ACs for their careful reading and acknowledge the unusual process this year, in which post-rebuttal discussion and score updates are constrained. Across all four reviews, we have directly addressed each substantive concern with additional experiments, clearer definitions, and sharper positioning, and the reviewers either explicitly confirm that their concerns are resolved or maintain reservations only about *framing* rather than correctness or methodological soundness. We respectfully ask the area chairs to evaluate the paper based on this updated and substantially strengthened record, which we believe now clearly supports an accept recommendation.

---

### Meta-Review · Area_Chair_fN9d · 2026-01-07

**Summary:**

After carefully checking the paper, the reviews, the rebuttal, and the author-reviewer discussions, two reviewers maintained a positive stance, while two reviewers remained negative. Both negative reviewers pointed out that the paper contains conceptual inconsistencies and imprecise definitions. After a closer re-examination of the manuscript, I find that these concerns have some merit. In particular, incorporating real human behavioral data for validation may further strengthen the contribution and improve the credibility of the work. Taken together, I recommend that the paper be revised and resubmitted after appropriate improvements.

**Reviewer Concerns:**

The score remains unchanged. I have read the rebuttal, and believe it does not improve the review. I believe the rebuttal has adequately addressed the concerns regarding experimental details and clarified several key definitions in the paper. However, it still does not sufficiently resolve the concerns about the rationale of the proposed framework, nor does it provide additional validation to justify its reasonableness.

**Reviewer Scores:**

One score may improve. Other score remains unchanged. I have read the rebuttal, and believe it will not affect the scores.

---

### Decision · Program_Chairs · 2026-01-26

Reject